# Killing by Type VI secretion drives genetic phase separation and correlates with increased cooperation

Luke McNally[1,2,*], Eryn Bernardy[3,*], Jacob Thomas[3], Arben Kalziqi[4], Jennifer Pentz[3], Sam P. Brown[3], Brian K. Hammer[3], Peter J. Yunker[4] & William C. Ratcliff[3]

By nature of their small size, dense growth and frequent need for extracellular metabolism, microbes face persistent public goods dilemmas. Genetic assortment is the only general solution stabilizing cooperation, but all known mechanisms structuring microbial populations depend on the availability of free space, an often unrealistic constraint. Here we describe a class of self-organization that operates within densely packed bacterial populations. Through mathematical modelling and experiments with *Vibrio cholerae,* we show how killing adjacent competitors via the Type VI secretion system (T6SS) precipitates phase separation via the 'Model A' universality class of order-disorder transition mediated by killing. We mathematically demonstrate that T6SS-mediated killing should favour the evolution of public goods cooperation, and empirically support this prediction using a phylogenetic comparative analysis. This work illustrates the twin role played by the T6SS, dealing death to local competitors while simultaneously creating conditions potentially favouring the evolution of cooperation with kin.

[1] Centre for Immunity, Infection and Evolution, School of Biological Sciences, University of Edinburgh, Edinburgh EH9 3FL, UK. [2] Institute of Evolutionary Biology, School of Biological Sciences, University of Edinburgh, Edinburgh EH9 3FL, UK. [3] School of Biological Sciences, Georgia Institute of Technology. Atlanta, Georgia 30332, USA. [4] School of Physics, Georgia Institute of Technology. Atlanta, Georgia 30332, USA. * These authors contributed equally to this work. Correspondence and requests for materials should be addressed to P.J.Y. (email: peter.yunker@physics.gatech.edu) or to W.R. (email: ratcliff@gatech.edu).

Microbes are fundamentally social organisms[1–5]. They often live in dense, surface-attached communities, and participate in a range of social behaviours mediated through the production and consumption of extracellular proteins and metabolites. Paradigmatic examples include the cooperative production of digestive enzymes[6], metal chelators[7], signalling molecules[6] and the structural components of biofilms[8]. Many of these extracellular compounds are susceptible to social exploitation, in which non-producing 'cheats' gain an evolutionary advantage. If unchecked, this social exploitation can lead to the extinction of cooperative genotypes[9,10].

It is widely recognized that the spatial segregation of cooperative microbes away from cheats can solve this cooperative dilemma by ensuring that the investment of cooperators goes to other adjacent cooperative individuals[1,5,10–12]. Mechanisms creating assortment when organisms expand their ranges via growth into free space have recently received much attention[13–18], where robust patterns of genetic segregation can occur via stochastic bottlenecking. However, this mechanism cannot generate genetic segregation within dense, well-mixed communities displaying no net growth, despite the clear ecological relevance of such communities.

One mechanism that has been proposed to potentially generate spatial structure in dense communities is antagonistic interactions among genotypes[1,19–23]. If different genotypes interact antagonistically then wherever a genotype is in the minority they will be killed by competitors at a high rate, resulting in genetically homogenous patches. While mechanisms via which individuals can recognize and kill non-kin have been extensively studied, the consequences of such interactions for the spatial structure of communities have not been explored in detail.

The Type VI secretion system (T6SS) is a potent mechanism of bacterial aggression that can deliver effector proteins directly into eukaryotic cells to mediate virulence by cellular disruption, and into adjacent bacteria to mediate competition by killing non-kin while leaving kin with corresponding protective immunity proteins unscathed[24,25]. In *Vibrio cholerae,* T6-proficient strains utilize the T6SS to intoxicate T6-deficient eukaryotic predators and diverse proteobacteria, as well as other more closely related *V. cholerae* isolates that lack identical effector immunity pairs[26–32]. T6-mediated segregation occurs during co-culture of T6-proficient *V. cholerae* with T6-deficient *E. coli*. Segregation was also predicted to occur between two mutually antagonistic T6-proficient strains[33], and recently demonstrated at the single cell level in co-cultures of *V. cholerae* and *Aeromonas hydrophila*[20].

Here we examine the causes and consequences of neighbour killing via the T6SS on the physical structure of microbial communities. Using a *V. cholerae* experimental system and mathematical modelling, we show that T6SS-mediated killing causes an initially well-mixed population of mutually antagonistic bacteria to phase separate, forming clonal patches that grow larger through time. This phase separation belongs to the 'Model A' class of order-disorder transitions, which is described by the Allen-Cahn equation. We mathematically demonstrate that the spatial structure generated as a consequence of T6SS-mediated killing can favour the evolution of public-goods cooperation by limiting the potential for unrelated 'cheats' to access secreted products. Finally, we bioinformatically show that bacteria with more T6SS systems and effectors dedicate a larger fraction of their genomes to secreted products. While it is too early to rule out alternative hypotheses, this correlation is consistent with general predictions from social evolutionary theory that spatially structured environments favour the evolution of cooperation.

## Results

**Mutual antagonism drives phase separation**. Our system illustrates the profound effect of T6SS-mediated killing on emergent spatial patterning of a surface attached population. Mathematical modelling suggests that an initially well-mixed population of mutual killers should rapidly undergo phase separation due to 'selfish herd' dynamics[34], as the cells within genetically uniform groups no longer risk T6SS-mediated death. Indeed, we observe rapid phase separation in three distinct classes of models, all starting with a randomly seeded population on a two-dimensional lattice (Fig. 1a). We first developed an individual-based model (IBM; Fig. 1b; Supplementary Movie 1) that simulates bacterial growth, the killing of adjacent competitors and reproduction into empty patches through time. IBMs are appealing, in that they offer an intuitive simulation of discretized, interacting individuals. However, IBMs often lack mathematical transparency, limiting generalization. We thus modelled our system using two distinct, mathematically defined approaches: an ecologically based partial differential equation model in order to gain analytical insight into the dynamics (PDE; Fig. 1c; Supplementary Figs 1 and 2, Supplementary Movie 2), and the ecologically-based Ising spin model in order to relate our results to classical modelling of phase separation in statistical mechanics[35] (Fig. 1d; Supplementary Movie 3). In all three modelling frameworks, initially well mixed populations rapidly underwent phase separation. Similarly, initially-well mixed populations of two *Vibrio cholerae* strains (C6706 and 692–79; Supplementary Table 1) capable of mutual T6SS-mediated killing (Supplementary Fig. 3) underwent phase separation (Fig. 1f,i,j). Non-killing controls (Δ*vasK,* that is, T6SS⁻; Supplementary Fig. 3) and T6SS⁺ mutual killers cultured at low temperatures which impede T6SS activity[36] remained well-mixed (Fig. 1e,g,h).

**Spatial analysis**. To determine whether our models and experiments undergo the same type of order-disorder transition, we quantitatively examined the dynamics of phase separation in each. We first computed the Fourier-transformed structure factor, $S(q)$. The characteristic wavenumber of clonal groups is $q_m = \int q S(q) dq / \int S(q) dq$, and the height of the peak is related to how often it occurs in the lattice (that is, the strength of patterning at that length scale). At early timesteps (Fig. 2a, Supplementary Fig. 4), or for non-killing controls (Fig. 2b), $S(q)$ is relatively flat, as expected for a well-mixed population lacking a characteristic length scale. T6SS-mediated killing causes $S(q)$ to increase at smaller values of $q$ (longer length scales) as the population grows increasingly structured. This progression of $S(q)$ is a hallmark of phase separation[37]. For Model A, $q_m$ scales as $q_m \propto 1/\sqrt{t}$, while $S(q_m)$ scales as $S(q_m) \propto t$ (ref. 38). It is ambiguous how to relate simulation time to experimental time; instead, we plot $S(q_m)$ versus $q_m$. All models (IBM, PDE and Ising) and experiments fall on the same line $(S(q_m) \propto 1/q_m^2)$ (Fig. 2c), a relationship consistent with the 'Model A' order-disorder phase separation process[39], developed to explain the interaction of atomic 'spins' in systems that lack conservation, and described by the Allen-Cahn equation $\frac{\partial \varphi}{\partial \tau} = \eta - \mu$, which relates the change in local concentration, $\varphi$, over time to diffusion and the chemical potential, $\mu$ and stochastic fluctuations (see Methods)[40]. To demonstrate this equivalence across wavenumbers, we plot $q_m^2 S(q)$ versus $q/q_m$ (Fig. 2d). This collapses all data onto one master curve. In fact, due to the universality of non-conserved domain growth, this collapse could have been expected. Importantly, this universality, shown in Fig. 2c,d, demonstrates that while initial conditions—such as the initial number ratio of the two competing strains—may influence the timing of phase separation, they do not influence how phase

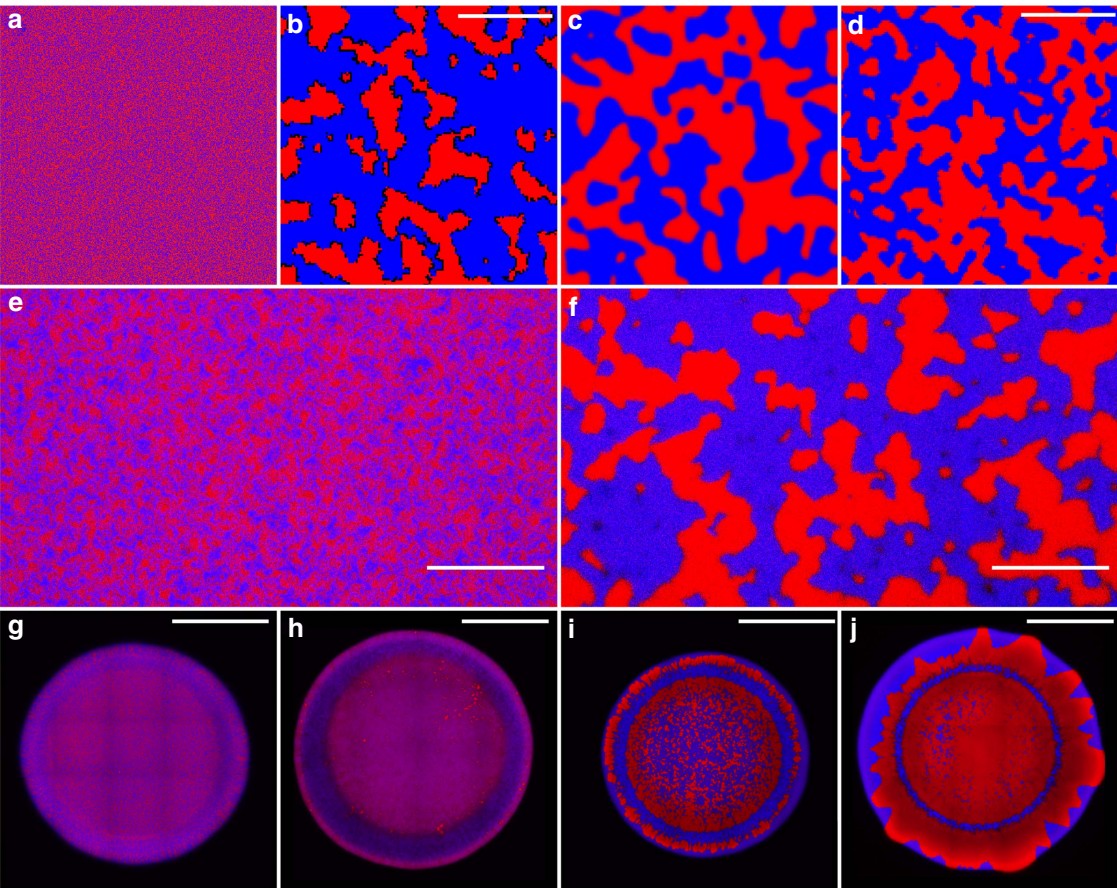

**Figure 1 | T6SS-mediated killing drives phase separation in dense bacterial populations.** We modelled the dynamics of phase separation in fully occupied, randomly seeded square lattices (**a**). Phase separation between red and blue bacteria capable of mutual killing occurred in an individual-based model (scale bar, 50 cells) (**b**), in a partial differential equation model (**c**), and in an Ising spin model (scale bar, 50 magnets) (**d**). No phase separation occurred between red (C6706) and blue (692–79) T6SS⁻ mutants of *Vibrio cholerae* (Δ*vasK*; **e**), in contrast to T6SS⁺ strains (**f**). We varied the efficacy of T6SS while still allowing for growth by culturing *V. cholerae* at a range of temperatures: 17 °C (**h**), 25 °C (**i**), and 30 °C (**j**). T6SS⁻ controls cultured at 25 °C did not phase separate (**g**). Scale bars denote 100 μm in **e**,**f**, and 1 mm in **g**–**j**. Images shown in **g**–**j** are representative of four replicate competitions.

separation occurs, or that the characteristic clonal group size always grows as $\sqrt{t}$. Cellular mobility has a surprising effect on phase separation: rather than impeding phase separation, it accelerates it by enhancing killing at the borders of clonal patches (Supplementary Methods, Supplementary Fig. 5 and Supplementary Movie 4).

To provide biological context for this process of phase separation, we calculated clonal assortment ($r$), for the IBM (Fig. 2e) and the *Vibrio* experiments (Fig. 2f). Assortment, which can be thought of as analogous to Hamiltonian relatedness[17,41,42], describes the extent to which clonemates spatially co-localize after accounting for their frequency in the population (see Methods for details). T6SS-mediated killing resulted in the creation of highly structured populations with high assortment over long length scales (Fig. 2e,f). Such assortment can protect diffusible public goods from consumption by competing strains[43,44].

**Spatial assortment supports cooperation.** To explore the effect of T6SS-mediated killing on the evolutionary stability of public goods cooperation, we introduced a diffusible cooperative good into our model. Because all three of our modelling frameworks displayed similar dynamics, we chose the PDE framework because it is the most amenable to analytical investigation. We considered two competing strains: a cooperator that secretes an exoproduct into its environment at an individual cost, and a

non-producing cheat that, all else equal, grows faster than the cooperator as it does not pay the cost of production. In this model, cellular growth rates for both strains depend on the local concentration of the diffusible exoproduct. We find that T6SS-mediated killing protects cooperation in two different ways. In a non-spatial (that is, constantly mixed) environment, T6SS-mediated killing can allow cooperators to resist invasion by rare cheats owing to the cooperators' numerical dominance in antagonistic interactions (that is, it creates positive frequency-dependence (Fig. 3c), while without T6SS-mediated killing (either because strains lack T6SS, or because the cheat is of the same T6SS type as the cooperator) cheats outcompete cooperators at all starting frequencies (Fig. 3a). However, in a spatially defined environment, phase separation driven by T6SS-mediated killing physically separates producers from cheats, expanding the conditions favouring cooperation and allowing them to invade a population of cheats from rarity (Fig. 3d,e and Supplementary Movie 5; see proof in Supplementary Methods), while cheats ultimately win in the absence of T6SS-mediated killing (Fig. 3b).

Our models and experiments demonstrate that T6SS-mediated killing can generate favourable conditions for the evolution of public-goods cooperation[1,12,15,45]. This can occur in two ways. First, T6SS-mediated killing induces positive frequency-dependent selection, allowing cooperators to resist rare cheats. Second, T6SS-mediated killing precipitates self-organized structuring of

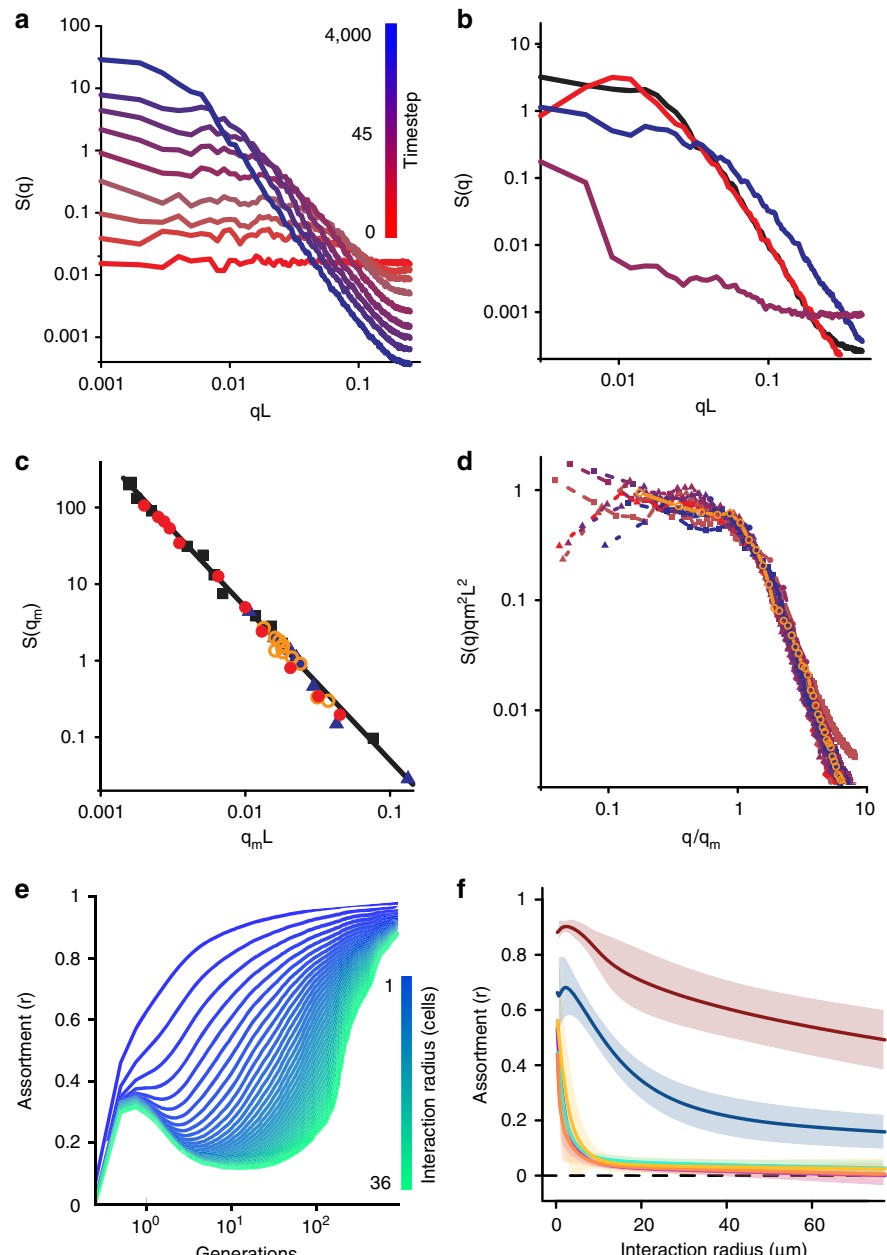

**Figure 2 | Structural analysis of models and experiments.** The static structure factor $S(q)$, plotted versus wavenumber $q$ multiplied by cell size $L$ for the individual based model (IBM; **a**) and for experiments (**b**). In the latter, the red and black lines depict two separate fields of view of *V. cholerae* strains C6706 and 692–79, started at an initial ratio of 1:6, while blue indicates a 1:8 inoculation ratio. The brown line depicts T6SS $^-$ mutants, and purple indicates mutual killers grown at 17 °C for 24 h (all others grown at 25 °C). (The brown line is obscured by the purple line, which is nearly identical.) Mutual killing drives phase separation, increasing $S(q)$ at smaller values of $q$. The relationship between $S(q_m)$ and $q_m$ is summarized in **c** with open orange cirlcles = experimental data (25 °C and a 1:6 inoculation ratio, as in **b**), black closed squares = IBM, red closed circles = PDE model ($d = 0.01$), and blue closed triangles = Ising model ($T = 1$); all three models and the experiments follow a universal $q_m^{-2}$ trend. $S(q)$ curves collapse when $S(q)q_m^2L^2$ is plotted versus $q/q_m$ (**d**), indicating that all models and experiments are undergoing the same coarsening process. Colour denotes model timestep, as in **a**, while symbols indicate type of model or experiment, as in **c**. We also examine the creation of spatial structure by calculating a biological metric, assortment ($r$), through time across 6,000 updates of the IBM (**e**) and after 24 h in experiments (**f**). Mutual killers were grown at 30 °C (red), 25 °C (blue) and 17 °C (green). Defective killers were grown at 30 °C (purple), 25 °C (teal) and 17 °C (orange). Plotted is the mean assortment of four replicate populations (mutual killers) and three replicate populations (defective killers) ± 95% confidence intervals.

microbial communities, reducing the diversity of cheats that can exploit cooperators. In addition to the predictions of our models previous experimental work has suggested that, via pleiotropic linkage to quorum sensing communication systems, the T6SS can also act as a policing mechanism protecting against the evolution of quorum sensing cheats[46]. Altogether, this suggests that there

are three complementary paths via which T6SS-mediated killing could favour the evolution of cooperation.

**Association between T6SS and secreted product evolution.** Does T6SS-mediated killing have a similar effect in the real world,

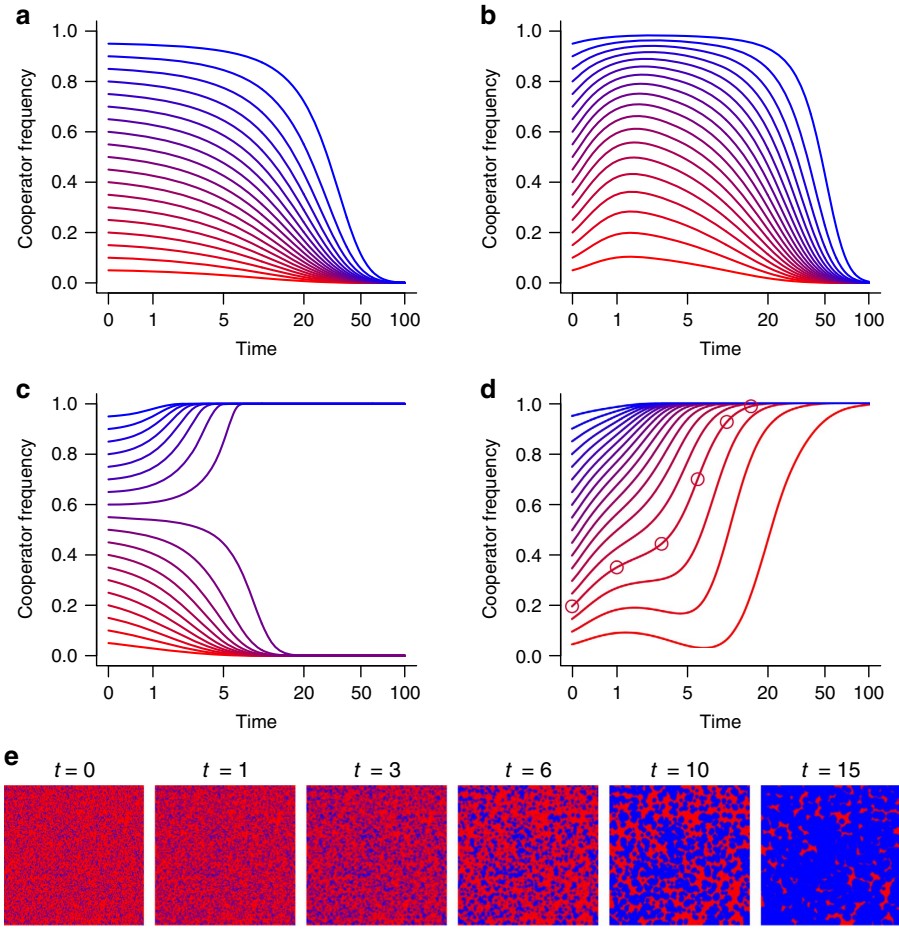

**Figure 3 | Phase separation favours the evolution of cooperation.** The dynamics of competition between cooperators and cheats are shown through time for different starting frequencies. In the absence of T6SS-mediated killing, cooperation is not favoured in either a well-mixed environment (**a**) or a spatially defined environment (**b**). In a non-spatial environment with killing via T6SS, cooperators can be protected from cheats when common owing to their advantage in antagonistic interactions, but cannot invade from rarity (**c**). In contrast, the high assortment created by phase separation allows cooperators to invade from rarity and spread to fixation (**d**). In **a–d**, line colour denotes initial cooperator frequency. The spatial organization of cooperators (blue) and cheats (red) during competition is shown in **e**. Panels correspond to the time-points marked by circles in **d**.

where ephemeral resources, physical disturbance and intense competition may impede these mechanisms? We approach this question phylogenetically, examining the relationship between the proportion of each genome coding for potentially exploitable secreted proteins and its T6SS complexity, with the rationale that microbes possessing a greater number of T6SSs may face less social exploitation by living in more structured communities. All else equal, genotypes that possess a greater number of T6SSs should form more highly structured patches (higher $r$), phase separating with a greater proportion of competitor genotypes (that is, those with non-complementary effector/immunity proteins). As a result, we hypothesize that genotypes with more T6SSs should experience less pressure from social cheating imposed by distantly related competitors. There is, of course, an important caveat to this scenario: while T6SS-based phase separation should effectively exclude competitors, it does not address the *de novo* evolution of cheating from within the clone (for example, Fig. 3a,b).

As a first-order proxy for cooperativity, we measured the proportion of a clone's genome dedicated to secreted proteins (henceforth referred to as 'secretome size'). While many of these secretions may have antagonistic effects on other microbes, they can still be seen as cooperative from the producing cell's perspective, as their kin can benefit from the reduced competition

that they create[47,48]. We constructed a Bayesian phylogenetic mixed model of T6SS-containing Proteobacteria and Bacteroidetes (Fig. 4a, Supplementary Fig. 6) using 439 genomes from 26 genera. Secretome size is positively correlated with both the number of T6SSs (Fig. 4b,d, Supplementary Table 2) and T6SS effector proteins (Fig. 4c,e, Supplementary Table 2) present, and the model shows an excellent overall fit to the data, explaining 99% of the variance in secretome size (Fig. 4f). These results are also robust in univariate analyses (Supplementary Tables 3 and 4) and to the inclusion of genome size as a predictor (Supplementary Table 5). As our analyses include many closely related strains (for example, many *Helicobacter pylori*, Fig. 4a), most (91%) of the variance in secretome size is explained by the phylogenetic relationships among strains. Nonetheless, the number of T6 secretion systems and T6SS effectors are important predictors of secretome size, explaining 8% of the total, and 90% of the non-phylogenetic variance in secretome size.

While the above analysis is consistent with the predictions of our mathematical model showing that phase separation should favour the evolution of cooperation (Fig. 3), it is not conclusive. As with any broad-scale phylogenetic analysis, alternative mechanisms explaining this correlation cannot be ruled out. For example, some unknown aspect of bacterial ecology may independently select for both investment in T6SS-mediated

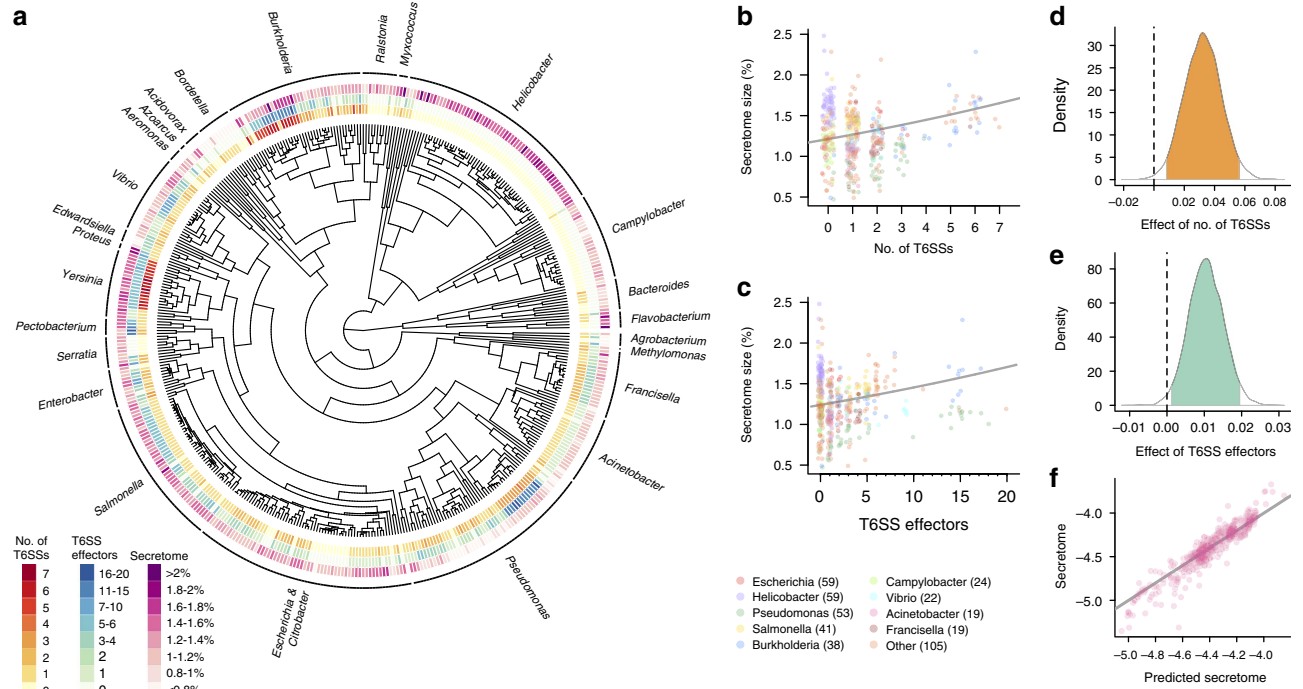

**Figure 4 | T6SS is associated with investment in other secreted products.** The phylogenetic distribution of T6SS, T6SS effectors and secretome size across 439 genomes from the Proteobacteria and Bacteroidetes (**a**). Secretome size of a strain (expressed as a percentage of genome size) increases with both its number of T6SSs (**b**) and T6SS effectors (**c**). Lines are the fits of univariate Bayesian phylogenetic mixed models (BPMMs) (Supplementary Tables 3 and 4). Posterior distributions of the effects of the numbers of T6SS (**d**) and T6SS effectors (**e**) on secretome size from the multivariate BPMM (Supplementary Table 2). Ninety-five per cent credible intervals of the estimates are shaded. Plot of observed against predicted secretome size from the multivariate BPMM (**f**), including effects of the number of T6SS, number of T6SS effectors and phylogeny. The line represents a 1:1 mapping.

killing and exoproduct production. By controlling for phylogenetic variation, our analysis should capture some of this ecological variance, through it does not eliminate it entirely. Alternatively, increases in the presence of dead competitors owing to T6SS-mediated killing can create additional opportunities for horizontal gene transfer[49], which could allow for increased acquisition of genes coding for secretions. However, such an explanation relies on horizontal gene transfer being biased towards genes encoding extracellular secretions. While there is evidence that this bias towards secretions is the case for plasmids and other mobile genetic elements[50], whether this bias occurs when sampling the genes of dead competitors remains to be determined. Detailed analysis of the phylogenetic dynamics of T6SS genes and secretions in individual clades may allow the relative contributions of these hypotheses to our observed correlation to be disentangled.

## Discussion

Phase separation is well-known to drive pattern formation in biology[51,52], but has mainly been investigated using either Turing activator-inhibitor feedbacks[53,54], or positive density-dependent movement, described by the Cahn-Hilliard equation[51,55–57]. In this paper, we describe a third general mechanism of self-organized pattern formation: targeted killing of non-kin competitors. This drives a 'Model A' phase separation; the kinetics of this coarsening process—described by the Allen-Cahn equation—only depend on a few cellular details. While we restrict our analysis in this paper to the T6SS, the role of antagonistic interactions in structuring biological communities it is probably far more general, applying to diffusible compounds that kill adjacent non-kin in both micro-organisms (for example, antibiotics) and macro-organisms (for example, allelopathy in plants[58] and animals[59]). However, while 'Model A' coarsening is universal, the realization of such dynamics in a densely packed, immobile, athermal system is likely unique to biology.

Physically, this system bears similarities to active matter[18,51,56,57]; phase separation has also been observed in these far from equilibrium active systems, wherein constituents expend energy to move. Phase separation in these systems typically occurs due to differences in mobility as a function of density; constituents move slowly through crowded regions, and quickly through low density regions. Mobility-induced phase separation has been observed (or predicted) in systems as varied as swimming bacteria[60], self-propelled colloids[61,62], mussels[51], granular rods[63], active filaments[64,65], rotating particles[66], among other systems[52]. In the current system, activity is derived from reproduction and killing events at high density rather than constituent mobility[67], leading to a 'Model A' transition.

Model A coarsening captures the behaviour of a broad array of phase transitions that lack conservation. This transition was originally developed to model magnetization in ferromagnetic materials via the Ising model. Ferromagnetic spins have minimum energy when they align; they do so via Glauber spin flips, leading to a change in the overall magnetization. The physical universality of this transition may be reflected in the strong correlation between secretome and T6SS effectors and apparatuses seen in Fig. 4. The microscopic details of the system do not strongly affect coarsening, so long as densely packed cells are equipped with T6SS.

In recent years, there has been a growing appreciation that many microbial behaviours requiring extracellular metabolism are susceptible to social exploitation. Here we show how simple cell-cell aggression can, as a consequence, create a structured

population favourable to cooperation. Clearly, many factors contribute to the structure and function of microbial communities[1,4,15,16,19–22,42,47]. However, because T6SSs are common (found in ∼ 25% of Gram-negative bacteria[68]), and microbes often live in dense communities, phase-separation driven by contact-mediated killing may have a fundamental role in defining the genetic composition and ecosystem-level functionality of microbial communities worldwide.

## Methods

**Bacterial strains and culture conditions.** The bacterial strains used in this study are shown in Supplementary Table 1. *Vibrio cholerae* fluorescence reporter constructs were chromosomally integrated and gene deletions and promoter replacements were constructed by allelic exchange as described and verified by Sanger sequencing[69–71]. *Vibrio cholerae* was routinely grown at 30 °C in Luria-Bertani (LB) liquid medium supplemented with 50 µg ml$^{-1}$ of kanamycin or 150 µg ml$^{-1}$ spectinomycin when appropriate. For confocal microscopy experiments, overnight cultures were mixed and 0.5 µl was inoculated onto Luria-Bertani agar (1.5%) pads on glass slides, and incubated at 17, 25 or 30 °C for 24 h. C6706 and 692–79 were inoculated at a 1:6 initial ratio, as T6SS$^{+}$ C6706 is more competitive than T6SS$^{+}$ 692–79 under our assay conditions (it grew from an inoculation ratio of 16.6% to constitute an average of 40–62% of each colony at all three temperatures). To visualize less-advanced stages of phase separation, we used a 1:8 initial ratio of strain C6706 to 692–79. For all images, we show C6706 in red and 692–79 in blue. As expected[69], phase occurred similarly when the fluorescence reporters were swapped between strains (Supplementary Fig. 7).

**Microscopy and image analysis.** Laser fluorescence confocal microscopy was performed with a Nikon A1R. The filters used were fluorescein isothiocyanate (for detecting mTFP1, cyan) and tetramethylrhodamine (TRITC) (for detecting mKO, orange). Full colony images were captured in one z-plane using the 20 × Plan Apo objective lens and a 2 × internal multiplier was applied to capture close-up images. The Galvano scanner was used to scan 2,048 × 2,048 pixels on all images in order to maximize resolution. For every sample, the top and bottom of the colony was located, and a plane in the middle was imaged. The images were stitched and channels were merged using NIS Elements software. To eliminate issues with red–green colourblindness, we present green fluorescence in images as blue.

To calculate the structure factor, $S(q)$, we start with an image from a simulation or experiment, $I(x,y)$. $S(q_x,q_y)$ is the absolute value of the Fourier Transform of $I(x,y)$, squared $S(q_x, q_y) = \left| \int I(x, y) e^{-2\pi \vec{q} \cdot \vec{r}} dx dy \right|^2$, where $q_x$ and $q_y$ are spatial frequencies in the x- and y- directions, respectively. We then radially average $S(q_x, q_y)$, $S(q) = \int S(q_x, q_y) d\theta$.

To calculate the assortment ($r$) of the genotype over interaction radius $h$, we again start out with a binarized image from a simulation or experiment $I(x,y)$ in which we set values of the focal strain $g$ to 1 and the competitor strain $c$ to $-1$. We first convolved $I(x,y)$ with a kernel in which all positions other than the center were set to 1, and the center set to $-((2h+1)^2 - 1)$, generating the transformed matrix $C(x,y)$. For example, the kernel for distance $h$ of 1 would be

$$\begin{array}{ccc} 1 & 1 & 1 \\ 1 & -8 & 1 \\ 1 & 1 & 1 \end{array}$$

Edges within distance $h$ were trimmed. For each interaction radius $h$ (which ranged from 1–36), we calculated the assortment $r$ of the focal strain $g$ as $r_g = (1 - \frac{C(x,y)_g}{2((2h+1)^2 - 1)} - \bar{g}) / (1 - \bar{g})$. $r$ is thus the mean frequency of $g$ within interaction radius $h$, relative to frequency of $g$ in the population as a whole. $r$, which ranges from $-1$ to 1, describes the spatial association of each genotype above or below what would be expected from random associations ($r = 0$). This definition of assortment is commonly used in social evolution studies, and is conceptually analogous to Hamiltonian relatedness[17,41,42]. Similarly, we calculated the assortment of the competitor strain $c$ as $r_c = (1 - \frac{C(x,y)_c}{2((2h+1)^2 - 1)} - \bar{c}) / (1 - \bar{c})$.

**Phylogenetic comparative analysis.** We gathered data on the presence of putative T6SSs and effectors across proteobacterial genomes from the SecReT6 database[72]. We restricted our analysis to genera in which there has been experimental verification of the presence of at least one T6SS in at least one strain in SecReT6 (ref. 72). This gave data for a total of 439 genomes from the Proterobacteria and Bacteroidetes of the genera *Acidovorax* ($N=5$), *Acinetobacter* ($N=19$), *Aeromonas* ($N=4$), *Agrobacterium* ($N=4$), *Azoarcus* ($N=2$), *Bacteroides* ($N=9$), *Bordetella* ($N=10$), *Burkholderia* ($N=10$), *Campylobacter* ($N=24$), *Citrobacter* ($N=2$), *Edwardsiella* ($N=4$), *Enterobacter* ($N=11$), *Escherichia* ($N=59$), *Flavobacterium* ($N=5$), *Francisella* ($N=19$), *Helicobacter* ($N=59$), *Methylomonas* ($N=1$), *Myxococcus* ($N=3$), *Pectobacterium* ($N=5$), *Proteus* ($N=2$), *Pseudomonas* ($N=53$), *Ralstonia* ($N=10$), *Salmonella* ($N=41$), *Serratia* ($N=9$), *Vibrio* ($N=22$) and *Yersinia* ($N=19$). For each genome we also recorded the genome size and secretome size (number of genes coding for secreted proteins) from

PSORTdb[73]. Any T6SS effectors identified in SecReT6 were removed from secretome size counts from PSORTdb to avoid creating a spurious correlation owing to double counting of effectors. Scatterplots of all raw data are shown in Supplementary Fig. 6. To control for the phylogenetic relationships among strains we used the SUPERFAMILY phylogeny[74], which we ultrametricised using the chronpl function in ape[75]. The data and phylogeny used are included in Supplementary Data 1 and 2.

We used a Bayesian phylogenetic mixed model (BPMM) approach to test for an evolutionary association between T6SSs and secretome size. Analyses were implemented in R using the package MCMCglmm[76]. We treated secretome size as a binomial response variable, expressing it as a proportion of genome size. In all models, we included phylogeny as a random effect to control for the shared evolutionary history of strains, and also included a residual random effect to account for overdispersion. For fixed effects we used an uninformative normally distributed prior with mean 0 and variance of $10^8$. For the phylogenetic and residual variances, we used an uninformative inverse gamma prior with shape and scale both set to 0.001. We ran all models for 6,000,000 iterations with a burn-in of 1,000,000, and thinning interval of 1,000 iterations. We used visual inspection of traces, as well as the Gelman-Rubin test[77,78] on three independent chains to assess model convergence. In all cases, the potential scale reduction factor was < 1.03. We first fit a model including both the number of T6SSs and number of T6SS effectors as fixed effects (Supplementary Table 2). To test the sensitivity of our results we also fit univariate models with number of T6SSs and number of T6SS effectors as fixed effects in isolation (Supplementary Tables 3 and 4). Finally, to control for the potential of non-linear scaling of secretome size with genome size we ran a model with number of T6SSs, number of T6SS effectors, and the log of genome size as fixed effects (Supplementary Table 5). In all cases, both the numbers of T6SSs and T6SS effectors show significant associations with secretome size. Statistics quoted are posterior modes, 95% credible intervals, and $p_{MCMC}$ a Bayesian equivalent to the frequentist *P* value, which is set as twice whichever is smaller of the proportion of posterior samples above or below zero. Values for proportion of variance explained (R-squared) were calculated following the approach of Nakagawa and Schielzeth[79] as the proportion of total variance attributable to the variable in question on the link (logit) scale, but removing the term for the intrinsic variance of the binomial distribution as we are interested in prediction at the level of proportion of the genome dedicated to secretions, rather than prediction of whether individual genes code for secretions. Total variance was calculated as the posterior mode of the sum of the residual, phylogenetic and fixed effect variance, with the proportional contributions of each component then determined. In addition, we calculated the proportion of non-phylogenetic variance accounted for by the fixed effects by repeating the calculations while neglecting the phylogenetic variance estimates.

**Individual based simulation model.** We randomly seeded a 500 × 500 lattice with an equal number of red and blue cells. Every time step, 5% of the cells were randomly chosen to activate their T6SS systems, killing any adjacent (eight cells surrounding the focal cell) cells of the opposite colour. Similarly, 5% of the cells in the landscape were randomly chosen to attempt to reproduce, filling up to one adjacent unoccupied patch with a cell of its colour. Rates of killing and reproduction were chosen to provide sufficient temporal resolution of population dynamics while still being computationally efficient. Reproduction was aborted if all neighbouring patches were occupied. Within each time step, model updates were propagated sequentially across rows, starting with the first position in the upper left corner. This model was coded in Python and is available upon request.

**Ising spin model.** We randomly seeded a 500 × 500 lattice with an equal number of '1' and '−1' entries, representing 'up' and 'down' magnetic spins. The eight nearest neighbours of each point in the lattice are summed and multiplied by the entry they circumscribe—if an entry is surrounded by neighbours of its own type, the resultant quantity (the Hamiltonian '*H*') is 8; if surrounded by neighbours opposite its own type, −8. We randomly choose a spin in the lattice and calculate $H_{old}$ and $H_{new}$, where $H_{new}$ is the Hamiltonian if the chosen spin were to flip, and $H_{old}$ the Hamiltonian if the spin were to remain unflipped.

Each spin flips independently, so flipping one spin does not change the sum of its eight nearest neighbours. Because the Hamiltonian for each entry, however, is the product of that sum with the entry itself, the net effect is simply to add a minus sign. Thus, the difference between the energy for the flipped spin and the unflipped spin is $\Delta H = H_{new} - H_{old} = (-H_{old}) - H_{old} = -2H_{old}$.

With this quantity calculated for our randomly chosen spin, we finally calculate $e^{-\Delta H} = e^{2H_{old}}$, and compare it to a random number between 0 and 1—if and only if it is larger, the spin will flip[80]. This process is applied iteratively, and after $500^2$ possible flips, one time-step is said to have passed. In the above discussion, the Hamiltonian is expressed in units of $k_B*T$. This proportionality can be adjusted to modify the speed of convergence to the phase-separated state; our simulation used a ratio of unity. No external field was applied. This model was coded in Mathematica and is available upon request.

**ODE model for well-mixed environment.** We will consider the interaction of two strains with different T6SS effector-immunity pairs so that each strain can kill the

other strain upon cell–cell contact. For simplicity, we assume that both strains differ only in their T6SSs, having the same basal growth rate $r$ and density dependent mortality rate $s$. We allow for asymmetric killing between the strains so that strain $A$ kills strain $B$ at rate $\alpha_{BA}$ and strain $B$ kills strain $A$ at rate $\alpha_{AB}$. From these assumptions, we can write the change in density of strains $A$ and $B$ in a well-mixed environment as

$$\frac{dA}{dt} = A(r - s(A+B) - \alpha_{AB}B) \quad (1a)$$

$$\frac{dB}{dt} = B(r - s(A+B) - \alpha_{BA}A). \quad (1b)$$

There are four possible equilibria for this system: a bacteria-free equilibrium at $A = 0$, $B = 0$; two single-strain equilibria at $A = r/s$, $B = 0$ and $A = 0$, $B = r/s$; and a coexistence equilibrium with both strains present at $A = \alpha_{AB}r/(s(\alpha_{AB} + \alpha_{BA}) + \alpha_{AB}\alpha_{BA})$, $B = \alpha_{BA}r/(s(\alpha_{AB} + \alpha_{BA}) + \alpha_{AB}\alpha_{BA})$. Note that at the coexistence equilibrium the ratio of strain $A$ to strain $B$ simply depends on their relative rates of killing ($A/B = \alpha_{AB}/\alpha_{BA}$). Analysing the Jacobian ($J$) of the system around the equilibrium we can see that the bacteria-free equilibrium is unstable as long as $r > 0$ ($\text{tr}(J) = 2r, |J| = r^2$), while the single strain equilibria are always stable for our assumption of positive killing ($\text{tr}(J) = -r(s + \alpha_{BA})/s$, $|J| = \alpha_{BA}r^2/s$ for $A = r/s$, $B = 0$, and $\text{tr}(J) = -r(s + \alpha_{AB})/s$, $|J| = \alpha_{AB}r^2/s$ for $A = 0$, $B = r/s$). Analysing the coexistence equilibrium we can see that both the trace and the determinant at the equilibrium are strictly negative

$$\text{tr}(J) = -\frac{\alpha_{AB}\alpha_{BA}rs}{s(\alpha_{AB} + \alpha_{BA}) + \alpha_{AB}\alpha_{BA}} \quad (2a)$$

$$|J| = -\frac{\alpha_{AB}\alpha_{BA}r^2}{s(\alpha_{AB} + \alpha_{BA}) + \alpha_{AB}\alpha_{BA}} \quad (2b)$$

meaning that this is a saddle node, and is hence unstable. This means that in a well-mixed environment killing by T6SS will always lead to one strain coming to dominate the environment, with the critical ratio of strains $A$ to $B$ at which $A$ eventually dominates decided by its relative rate of killing ($A$ dominates if $A/B > \alpha_{AB}/\alpha_{BA}$) and vice versa for domination by strain $B$ (Supplementary Fig. 1).

**PDE model for spatially extended environment.** In order to study the dynamics of strains with different T6SS effector-immunity pairs in a spatial environment, we use the following system of PDEs

$$\frac{\partial A}{\partial t} = A(r - s(A+B) - \alpha_{AB}B) + d\Delta A \quad (3a)$$

$$\frac{\partial B}{\partial t} = B(r - s(A+B) - \alpha_{BA}A) + d\Delta B. \quad (3b)$$

The dynamics are given as before but with a Laplacian operator for the diffusion/dispersal of cells through space, where $d$ is the dispersal/diffusion rate for both strains. We will first consider the stability of a homogenous coexistence equilibrium where the average densities of the strains are

$$A_0 = \frac{\alpha_{AB}r}{s(\alpha_{AB} + \alpha_{BA}) + \alpha_{AB}\alpha_{BA}} \quad (4a)$$

$$B_0 = \frac{\alpha_{BA}r}{s(\alpha_{AB} + \alpha_{BA}) + \alpha_{AB}\alpha_{BA}}. \quad (4b)$$

We will consider the effect of a fluctuation in the strain composition in one-dimensional space of the form

$$A(x) = a \sin \beta x + A_0 \quad (5a)$$

$$B(x) = -a \sin \beta x + B_0. \quad (5b)$$

This fluctuation leads to a change in density of strain $A$ at location $x$ of $a \sin \beta x$ with a corresponding change of $-a \sin \beta x$ in the density of strain $B$, where $a$ is the amplitude of the fluctuation (which e will assume to be infinitesimally small) and $\beta$ is the angular frequency. As we are concerned with changes in composition we will denote the density/volume fraction of strain $A$ as $\varphi_A = A/(A+B)$ and using equations (3a and 3b) the rate of change in the density fraction is

$$\frac{\partial \varphi_A}{\partial t} = \frac{B(d\Delta A - \alpha_{AB}AB) - A(d\Delta B - \alpha_{BA}AB)}{(A+B)^2}. \quad (6)$$

Substituting in equations (4a, 4b, 5a and 5b) we get

$$\frac{\partial \varphi_A}{\partial t} = -\frac{a \sin \beta x}{(\alpha_{AB} + \alpha_{BA})r^2}\left(r\left(\beta^2 d(s(\alpha_{AB} + \alpha_{BA}) + \alpha_{AB}\alpha_{BA})\right)\right.$$
$$+ a \sin \beta x(s(\alpha_{AB} + \alpha_{BA}) + \alpha_{AB}\alpha_{BA}) + (r(\alpha_{AB} - \alpha_{BA}) \quad (7)$$
$$\left. + a \sin \beta x(s(\alpha_{AB} + \alpha_{BA}) + \alpha_{AB}\alpha_{BA}))\right).$$

For the compositions to diverge $\partial \varphi_A/\partial t$ must have the same sign as the fluctuation $\sin \beta x$, as this leads to $A$ increase in frequency in regions of positive fluctuation and $B$ to increase in frequency in regions of negative fluctuation. At any point of zero compositional fluctuation ($x = n\pi/\beta$, where $n \in \mathbb{Z}$), $\partial \varphi_A/\partial t = 0$ and this condition cannot be fulfilled. However, the condition for $\partial \varphi_A/\partial t$ and $\sin \beta x$ to have the same

sign in a region of non-zero fluctuation ($x \neq n\pi/\beta$, where $n \in \mathbb{Z}$) is

$$\frac{\alpha_{AB}\alpha_{BA}r}{\alpha_{AB}\alpha_{BA} + s(\alpha_{AB} + \alpha_{BA})}$$
$$- \frac{a \sin \beta x(r(\alpha_{AB} - \alpha_{BA}) + a \sin \beta x(s(\alpha_{AB} + \alpha_{BA}) + \alpha_{AB}\alpha_{BA}))}{r} \quad (8)$$
$$> d\beta^2.$$

Assuming that the amplitude of the compositional fluctuation is infinitesimally small ($a \to 0$), inequality 8 simplifies to

$$\frac{\alpha_{AB}\alpha_{BA}r}{\alpha_{AB}\alpha_{BA} + s(\alpha_{AB} + \alpha_{BA})} > d\beta^2. \quad (9)$$

Therefore, the strain compositions will diverge, with $A$ increasingly dominating in regions of positive compositional fluctuation and $B$ coming to dominate in regions of negative compositional fluctuation, if the fluctuations occur on a sufficiently wide scale relative to bacterial dispersal/diffusion (that is, sufficiently low $d$ and/or $\beta$, Supplementary Fig. 2). It is also worth noting that this condition is more easily satisfied for higher rates of killing by both strains (higher $\alpha_{AB}$ and $\alpha_{AB}$), and for higher values of the basal growth rate (higher $r$) and lower values of density-dependent mortality (lower $s$). This highlights that the local demography of killing by T6SS pushes the system towards decomposition even for fluctuations on short length scales, while higher diffusion/dispersal of cells mean that fluctuations of longer length scales are required for decomposition to occur.

**Relationship of the PDE model to the Allen-Cahn equation.** The Allen-Cahn equation governs the density of strains A and B during phase separation absent the conservation of strains A and B. Assuming that the *effective* diffusion rate $d_{\text{Eff}}$ doesnot vary through space (as will be approximately the case when oscillations in composition are of vanishingly small amplitude, $a \to 0$) we can write the Allen-Cahn equation for strain A as

$$\frac{\partial \varphi_A}{\partial t} = \Gamma\left(d\Delta \varphi_A - \frac{\partial f(\varphi_A)}{\partial \varphi_A}\right). \quad (10)$$

Where $f(\varphi_A)$ is a function of $\varphi_A$. The $\partial f(\varphi_A)/\partial \varphi_A$ term drives $\varphi_A$ to the free energy minimum over time, and can work with or against diffusion. Thus, the effective diffusion is uphill whenever $\partial \varphi_A/\partial t$ and $\Delta \varphi_A$ are of opposite sign (that is, whenever the change in volume fraction is of opposite sign to the Laplacian). The Laplacian, $\Delta \varphi_A$ is given by

$$\Delta \varphi_A = -\sin \beta x \frac{a\beta^2(\alpha_{AB}\alpha_{BA} + s(\alpha_{AB} + \alpha_{BA}))}{r(\alpha_{AB} + \alpha_{BA})}. \quad (11)$$

The sign of $\Delta \varphi_A$ is, therefore, given by the sign of $-\sin \beta x$, and it follows that $\partial \varphi_A/\partial t$ and $\Delta \varphi_A$ will be of opposite sign, and effective diffusion will be uphill, in the limit of infinitesimally small amplitude fluctuations ($a \to 0$) whenever inequality 9 is satisfied. This highlights the two counteracting forces governing whether decomposition occurs—the diffusion/dispersal of cells favours downhill diffusion, while the local demographic effects of killing mimic uphill diffusion by a strain's relative growth rate being increased in regions in which it is in the majority.

**Adding public goods to the model.** To consider the effects of T6SS-mediated phase separation on the evolution of cooperation we extend our model so that strain A produces a diffusable public good secretion at rate $\rho$, while strain B does not invest in its production. We assume that strain A pays a growth rate cost $c$ for production of the secretion. We also assume that the secretion increases each strains growth rate by amount $b$ per unit concentration $S$ by increasing nutrient availability (for example, an exoenzyme digesting a substrate or siderophores binding insoluble iron). From these assumptions, we write the dynamics of the two strains and the secretion concentration as

$$\frac{dA}{dt} = A(r - c + bS - s(A+B) - \alpha_{AB}B) \quad (12a)$$

$$\frac{dB}{dt} = B(r + bS - s(A+B) - \alpha_{BA}A) \quad (12b)$$

$$\frac{dS}{dt} = \rho A - \lambda S \quad (12c)$$

for a well-mixed, non-spatial system and as

$$\frac{\partial A}{\partial t} = A(r - c + bS - s(A+B) - \alpha_{AB}B) + d\Delta A \quad (13a)$$

$$\frac{\partial B}{\partial t} = B(r + bS - s(A+B) - \alpha_{BA}A) + d\Delta B \quad (13b)$$

$$\frac{\partial S}{\partial t} = \rho A - \lambda S + D\Delta S \quad (13c)$$

for a spatially extended system, where $\lambda$ is the decay rate of the secretion, $D$ is its diffusion coefficient, and all other variables are as previously defined. We

numerically explore both the non-spatial and spatial systems in the presence ($\alpha_{AB} > 0, \alpha_{BA} > 0$) and absence ($\alpha_{AB} = 0, \alpha_{BA} = 0$) of killing in the Fig. 3, and here we analytically explore these four scenarios.

**Non-spatial model in the absence of killing.** In order to gain analytical insight into our model, we will make the assumption that the dynamics of the public good occur on a much faster time-scale than the ecological dynamics so that we may treat the public good as being at equilibrium for any given ecological state of the model. For the non-spatial model this gives us

$$S = \frac{\rho A}{\lambda}. \tag{14}$$

Substituting this into equation (10) and setting $\alpha_{AB} = 0, \alpha_{BA} = 0$ the dynamics of the two strains are

$$\frac{dA}{dt} = A\left(r - c + \frac{\rho bA}{\lambda} - s(A + B)\right) \tag{15a}$$

$$\frac{dB}{dt} = B\left(r + \frac{\rho bA}{\lambda} - s(A + B)\right). \tag{15b}$$

We will assume throughout that $s > \rho b/\lambda$, such that in the absence of cheaters a population of cooperators has a finite stable equilibrium. Solving for the equilibrium of equations (15a and 15b) we can see that the only stable equilibrium is $B = r/s$ so long as $c > 0$. This means that, as long as there is a cost of cooperation, cheaters will always outcompete cooperators in a non-spatial environment in the absence of killing.

**Non-spatial model with killing.** Again using the assumption that public goods dynamics play out on a faster time-scale than the ecological dynamics but now setting $\alpha_{AB} = \alpha_{BA} = \alpha$ (that is, assuming symmetric killing) the dynamics of the two strains are

$$\frac{dA}{dt} = A\left(r - c + \frac{\rho bA}{\lambda} - s(A + B) - \alpha B\right) \tag{16a}$$

$$\frac{dB}{dt} = B\left(r + \frac{\rho bA}{\lambda} - s(A + B) - \alpha A\right) \tag{16b}$$

For this system there are two single strain equilibria, one for a pure population of cooperators at

$$A = \frac{\lambda(r - c)}{\lambda s - \rho b} \tag{17a}$$

$$B = 0 \tag{17b}$$

and one for a pure population of cheaters at

$$A = 0 \tag{18a}$$

$$B = \frac{r}{s}. \tag{18b}$$

The cooperator equilibrium is stable so long as $\alpha > c(\lambda s - \rho b)/(\lambda r - \lambda c)$, while the cheater equilibrium is always stable. There is a an unstable saddle point at

$$A = \frac{\lambda(\alpha r + cs)}{\alpha(\alpha\lambda + 2\lambda s - \rho b)} \tag{19a}$$

$$B = \frac{\alpha\lambda r + \rho bc - c(\alpha + s)}{\alpha(\alpha\lambda + 2\lambda s - \rho b)} \tag{19b}$$

which exists whenever $\alpha > c(\lambda s - \rho b)/(\lambda r - \lambda c)$. This means that in the presence of killing cooperation can potentially still be a stable outcome. However, the location of the saddle point given is always at a higher value of $A$ than of $B$ (that is, $A$ is always larger than $B$ in equations (19a and 19b) whenever $c > 0$), meaning that the basin of attraction for strain $A$ is smaller than that for strain $B$. Thus, though killing can help protect cooperation from cheaters in a non-spatial environment, they are still disfavoured compared with cheaters. More generally, the condition for the cooperative strain to increase in frequency is

$$\alpha(A - B) > c. \tag{20}$$

Here we see that, in a non-spatial environment killing can help protect cooperators owing to the positive frequency dependence that it induces. However, cooperators are still disfavoured compared with cheaters as the basin of attraction for cheaters is larger (equations (19a and 19b)).

**Spatial model in the absence of killing.** We will consider a one-dimensional spatial environment with a fluctuation in the strain composition as given in equations (5a and 5b). Using this spatial distribution of strains and assuming that the public goods dynamics happen on a much faster timescale than the ecological dynamics, we can solve for the equilibrium concentration of the public good across

space as

$$S(x) = \frac{\rho A_0}{\lambda} + \frac{\rho a \sin\beta x}{\beta^2 D + \lambda}. \tag{21}$$

Note here that as the diffusion rate of the public good approaches infinity ($D \to \infty$) this simplifies to $S(x) = \rho A_0/\lambda$, and thus has a constant concentration through space, while as the diffusion rate of the public good approaches zero it simplifies to $S(x) = \rho(A_0 + a \sin\beta x)/\lambda$, and is thus simply proportional to the local density of the cooperator strain. Using equations (11, 18a and 18b) and setting $\alpha_{AB} = \alpha_{BA} = 0$ we get

$$\frac{\partial A}{\partial t} = (A_0 + a\sin\beta x)\left(r - c - s(A_0 + B_0) + \frac{\rho bA_0}{\lambda} + \frac{\rho ba\sin\beta x}{\beta^2 D + \lambda}\right) - \beta^2 ad\sin\beta x. \tag{22a}$$

$$\frac{\partial B}{\partial t} = (B_0 - a\sin\beta x)\left(r - s(A_0 + B_0) + \frac{\rho bA_0}{\lambda} + \frac{\rho ba\sin\beta x}{\beta^2 D + \lambda}\right) + \beta^2 ad\sin\beta x \tag{22b}$$

We can evaluate the fitness (per cell growth rate) of each strain as

$$w_A = \frac{\int \partial A/\partial t\ dx}{\int A\ dx} \tag{23a}$$

$$w_B = \frac{\int \partial B/\partial t\ dx}{\int B\ dx}. \tag{23b}$$

We can then evaluate when cooperators are favoured by evaluating the inequality $w_A > w_B$, which gives

$$\frac{\rho a^2 b(A_0 + B_0)}{2A_0 B_0\left(\beta^2 D + \lambda\right)} > c \tag{24}$$

as the condition for cooperators to increase in frequency. This condition is more easily favoured for lower costs of cooperation (low $c$), higher benefits of cooperation (high $b$), and with spatial fluctuations that are of large amplitude (high $a$) and over a wide spatial scale (low $\beta$).

Inequality 24 shows that cooperation can be favoured by spatial variance in the population composition. However, in the absence of outside forces or stochastic effects will such spatial fluctuations be maintained? To answer this, we first evaluate an expression for the change in the density/volume fraction of the cooperator strain giving

$$\frac{\partial \varphi_A}{\partial t} = -\frac{cA_0 B_0 - a\sin\beta x\left(c(A_0 - B_0) - \beta^2 d(A_0 + B_0) + ac\sin\beta x\right)}{(A_0 + B_0)^2} \tag{25}$$

which is always negative, meaning that the cooperator strain is always locally decreasing in frequency, even when globally increasing in frequency. This means that the change in cooperator frequency cannot match the sign of the spatial fluctuation $\Delta\varphi_A$, and thus in the absence of external forces any spatial fluctuations in composition will be lost. As we can see from inequality 24 as the amplitude of the fluctuation decays to zero ($a$ approaches 0) cooperation cannot be favoured, and thus without killing, external perturbations to the system or stochastic effects are required to maintain the spatial structure necessary to maintain cooperation.

**Spatial model with killing.** We now set $\alpha_{AB} = \alpha_{BA} = \alpha > 0$ and evaluate the consequences of the combination of killing and a spatial environment of the dynamics of cooperation. From our assumptions, the dynamics are now given by

$$\frac{\partial A}{\partial t} = (A_0 + a\sin\beta x)\left(r - c - s(A_0 + B_0) - \alpha(B_0 - a\sin\beta x) + \frac{\rho bA_0}{\lambda} + \frac{\rho ba\sin\beta x}{\beta^2 D + \lambda}\right) - \beta^2 ad\sin\beta x \tag{26a}$$

$$\frac{\partial B}{\partial t} = (B_0 - a\sin\beta x)\left(r - s(A_0 + B_0) - \alpha(A_0 + a\sin\beta x) + \frac{\rho bA_0}{\lambda} + \frac{\rho ba\sin\beta x}{\beta^2 D + \lambda}\right) + \beta^2 ad\sin\beta x. \tag{26b}$$

We can again evaluate the fitness of each strain as in equations (23a and 23b), and derive the condition for the cooperative strain to be favoured ($w_A > w_B$) as

$$\frac{\rho a^2 b(A_0 + B_0)}{2A_0 B_0\left(\beta^2 D + \lambda\right)} + \frac{\alpha(A_0 - B_0)(2A_0 B_0 - a^2)}{2A_0 B_0} > c. \tag{27}$$

Note that if the cooperator and cheater strains are at equal average density ($A_0 = B_0$) this simplifies to the inequality given in 24. As the amplitude of the spatial fluctuation must be less than the average density of the less abundant of the two strains ($a < \min\{A_0, B_0\}$) the condition given in inequality 27 is more easily satisfied than that in inequality 24 whenever $A_0 > B_0$ and less easily satisfied whenever $A_0 < B_0$. This occurs owing to the positive frequency dependence introduced by killing. However, as we will show, unlike in a spatial environment

without killing, killing in a spatial environment can increase structuring, thus further favouring cooperation.

Comparing inequality 27 (condition for cooperation to increase in frequency in a spatial environment with killing) with inequality 20 (condition for cooperation to increase in frequency in a non-spatial environment with killing), we can also see that the presence of spatial heterogeneity means that the cooperator can be favoured even when numerically less abundant in a spatial environment with killing (that is, inequality 27 can be satisfied when inequality 20 is not). This occurs as spatial heterogeneity can allow cooperators to disproportionately gain the benefits of cooperation compared with cheaters.

Finally, we will show that killing causes phase separation in a spatial environment when one strain is a cooperator and the other a cheat. We follow the same approach as before and consider whether a spatial fluctuation around the homogenous coexistence equilibrium given in equations (17a and 17b) will be amplified, which can be evaluated by considering if $\partial \varphi_A / \partial t$ has the same sign as the fluctuation $\sin \beta x$. At any point of zero compositional fluctuation ($x = n\pi/\beta$, where $n \in \mathbb{Z}$), $\partial \varphi_A / \partial t = 0$ and this condition cannot be fulfilled. However, the condition for $\partial \varphi_A / \partial t$ and $\sin \beta x$ to have the same sign in a region of non-zero fluctuation ($x \neq n\pi/\beta$, where $n \in \mathbb{Z}$) in the limit of an infinitesimally small amplitude for the compositional fluctuation ($a \rightarrow 0$) is

$$\frac{2\lambda(\alpha r + cs)(\alpha\lambda r - c(\alpha + s) + \rho bc)}{(\alpha\lambda(2r - c) + \rho bc)(\alpha\lambda + 2\lambda s - \rho b)} > d\beta^2. \qquad (28)$$

This shows that killing promotes phase separation of cooperator and cheater strains in a spatial environment which favours cooperators of cheats as shown in inequality 27.

Taken together, these results show that: (1) Killing in a non-spatial environment can protect cooperators from cheats when the cooperator is at higher abundance. However, cooperators cannot invade from rarity; (2) Heterogeneity in a spatial environment without killing can favour cooperators over cheats. However, in the absence of any external forces or stochastic effects (for example., bottlenecking during range expansion) this structure will ultimately be lost, allowing cheaters to win; and (3) In a spatial environment with killing phase separation can occur, protecting cooperators from cheats and potentially allowing them to invade from rarity.

**Parameter values for numerical simulations of PDE model.** Equations were numerically evaluated in R. Parameter values for figures and videos are as follows: Figs 1 and 2 and Supplementary Movie 2: $r = 2$, $s = 2$, $\alpha_{AB} = 0.5$, $\alpha_{BA} = 0.5$, $d = 0.01$; Supplementary Fig. 5: $r = 2$, $s = 2$, $\alpha_{AB} = 0.5$, $\alpha_{BA} = 0.5$; Supplementary Movie 4: $r = 2$, $s = 2$, $\alpha_{AB} = 0.5$, $\alpha_{BA} = 0.5$, and $d$ as indicated in the panels; Fig. 3 and Supplementary Movie 5: $r = 2$, $s = 2$, $\alpha_{AB} = 0.5$, $\alpha_{BA} = 0.5$, $d = 0.01$, $b = 1.9$, $c = 0.1$, $D = 0.1$, $\lambda = 100$, $\rho = 100$.

**Code availability.** All code is available from the authors upon request.

**Data availability.** All data sets generated or analysed during this study are included in this published article (and its Supplementary Information files) or are available from the corresponding authors upon request.

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

## Acknowledgements

This work was supported by NSF grant #DEB-1456652 to W.R. and NASA Exobiology grant #NNX15AR33G to W.R.; by the Gordon and Betty Moore Foundation grant #4308.07 to B.H., NSF grant #MCB-1149925 to B.H. and Wellcome Trust grant #WT095831 to S.B. L.M. was supported by HFSP grant #RGP0011/2014 to S.B. We would like to thank David Yanni, Jonathan Michel, Shane Jacobeen and Wouter Ellenbroek for helpful comments.

## Author contributions

All authors actively participated in the planning of this research. E.B., J.T. and B.H. constructed the *Vibrio* mutants and performed all bacterial experiments. E.B., J.T., P.J.Y. and A.K. performed and analysed the microscopy. L.M. and P.J.Y. wrote and analysed the PDE model. L.M. performed the phylogenetic analysis. W.R. wrote the individual-based model and script to calculate assortment, P.J.Y. and A.K. wrote the Ising spin model and script to calculate $S(q)$. W.R., L.M., E.B. and P.J.Y. wrote the first draft of the paper, all authors contributed to revision.

## Additional information

**Competing financial interests:** The authors declare no competing financial interests.

**Publisher's note**: 

