## [Peer Review File · Nature Communications]

Reviewers' comments:

Reviewer #1 (Remarks to the Author):

The manuscript "Killing by Type VI secretion drives clonal phase separation and the evolution of cooperation" by McNally et al studies a form of self-organization among bacteria that kill each other via the type VI secretion system (T6SS). The authors observe two bacterial strains that kill each other to form a pattern in various mathematical models side by side with an experiment. Analysis of the dynamics of the phase separation resulting in the pattern formation reveals a separation process between the two strains that can be described by the Allen-Cahn equation and differs from other described forms of phase separation in biology. Spatial organization of microbes affects a multitude of traits such as cooperation, which the authors further focus on and present a model of how phase separation by the T6SS could enable cooperators to take over populations in which they were mixed with cheats. Along the line of this prediction that phase separation by T6SS-mediated killing creates conditions in favour of cooperative traits, the authors lastly perform a broad analysis of Proteobacteria and Bacteroidetes and present a positive correlation between the size of the secretome, the number of T6SSs and the number of T6SS effectors of a given strain.

Spatial structure is key to the interactions between microbes. Understanding the dynamics of microbial self-organization underlying the spatial structure within a microbial community is therefore of high interest. McNally et al characterize the dynamics of how two sets of microbes that kill each other via the type VI secretion system assemble in a community. The findings advance our understanding of bacterial communities, microbial self-organization, T6SS-mediated killing and putative effects thereof on microbial interactions.

Overall, the manuscript presents interesting findings. The authors use an elegant approach and provide extensive documentation in supplementary files. I can see the results being of interest to a wide readership.

More detailed comments that should be addressed are listed below:

Major comments

- (title (1)) "clonal phase separation". I would be careful with the term "clonal" here. Experimental data from the T6SS field indicates that killing or no killing (immunity) only depends on the effector- and immunity protein, and is, roughly speaking, independent of the rest of the genome. In other words, two strains that are not clonal but share the same effector and immunity pairs would be immune and not separate. Although "clonal" is correct based on the way the experiments are set up in this manuscript, I recommend to at least remove "clonal" from the title and maybe somewhere in the manuscript refer to the concept that all that matters is the set of immunity genes/proteins independent of actual clonality over the whole genome.

- (title (2)) "drives ... the evolution of cooperation". Based on what is presented in this manuscript I think it is fair to conclude that T6SS killing drives phase separation but I think it is too much to conclude that T6SS-mediated killing drives the evolution of cooperation.

Overall, the manuscript should have an emphasis on the results of the first half that are validated through mathematical models and experiments.

- Overall, the manuscript is very short and condensed. Considering that the results might be of interest to a wide readership, I think it would benefit from a more detailed introduction, explanation of the rationale for choosing particular models and more detailed conclusion. For example, if one of the main findings of the paper is the description of the dynamics by which the phase separation happens, how does this dynamic differ from the other forms of phase separation (discussion) and why is phase separation within biological systems itself not introduced in the introduction with more details?

- T6SS-mediated killing among *V. cholerae* is used as an experimental model system in this manuscript. Experimental work has been performed on the T6SS of *V. cholerae* (Pukatzki et al., 2006, Basler et al., 2012), contact-dependent prokaryotic killing by *V. cholerae* via the T6SS (MacIntyre et al., 2010) and killing between *V. cholerae* strains based on the presence/absence of effector- and immunity genes (Dong et al. 2013, Unterweger et al., 2014) and should be cited. A study not using *V. cholerae* but *Burkholderia thailandensis* indicates that the T6SS might be used to get rid of cheats that do not contribute to quorum-sensing related costs (Majerczyk et al., 2016). This paper might be of interest to the authors and I think should be referred to in the context of T6SS and cooperative traits. Although the paper by Borenstein et al. is referenced, I recommend putting the results and the models used here a little more in the context of the results and models of Borenstein et al.

- Figure 2b shows how sensitive the phase separation described in here is to the starting ratios at which the two strains are mixed in an experiment. This should be mentioned and commented on in the text. Did the authors determine the actual ratio of the two strains at some point by plating out the mix and counting CFUs? Do I read this correctly in Figure S1 that the bacteria have been mixed at a 1:10 ratio in this experiment?

Minor comments

- l. 23 "novel" not necessary

- l. 28f I can see from the data presented later on how mathematic models predict that T6SS-mediated killing created spatial patterns that potentially favour cooperative traits. It is not clear to me, how the mathematical model makes the prediction about the "evolution" of such traits as referred to in this sentence. I also think that the results of the analysis shown in the last figure do not actually "verify" this prediction as phrased here.

- l. 31 based on what is presented in the manuscript, the final sentence of the abstract might rather read "...while simultaneously creating conditions potentially favoring cooperation with kin".

- l. 40ff "No mechanisms have been described...": I consider this a very bold statement. Maybe good to specify what "mechanism" refers to here. If "mechanism" refers to the biological mechanism of the T6SS, T6SS-mediated killing has been described in the past (see references mentioned above). What about formulating this sentence as a question, something like "Considering the ecological relevance of dense, well-mixed communities displaying no net growth, we were wondering if microbes are able to genetically segregate

in such communities.”

- l. 45ff, this sentence could be potentially misleading. Effectors are delivered to neighbouring cells independently of what they are. Maybe modification as follows in l. 46f would help “... proteins to adjacent cells, competitors die whereas clonemates with ...)
- l.47, re-consider references for this sentence. Publications for prokaryotic killing of *V. cholerae* and protection by effector-immunity interactions that could be cited here are: MacIntyre et al., 2010; Dong et al., 2013, Brooks et al., 2013. In more general terms of antiprokaryotic killing via the T6SS independent of the species, also the following papers could be cited: Hood et al., 2010, Russell et al., 2011 or reviews such as Ho et al., 2014; Cianfanelli et al., 2016
- l. 47, l.27 “new class of active matter”: the wording sounds a little too dramatic to me.
- l. 54 rationale for why these three models were chosen would be helpful.
- l. 62ff the variable q (wavelength) is described in the figure legend but not the next. The authors might consider adding a description of q to the text.
- l. 75 for a reader not familiar with the Model A phase separation process, it might be helpful providing more background, for example on other processes that are described with this equation
- l. 82 it is not clear why the IBM was chosen in Figure 2 among the three models used in Figure 1
- l. 82 ff “T6SS-mediated killing...”: The authors should consider disconnecting the experimental results in this paper from the results in the referenced papers by replacing the comma with a period. Based on the findings of the referenced papers, it is likely that the longer length scales protect diffusible public goods from cheats here but it has not been tested in this exact setting studied here.
 - l. 86: No rationale was mentioned why the PDE model is used here out of the three models that were used at the beginning.
 - l. 86ff: Has such a model of a cooperator and a non-producing cheat been used before in a different context and if so for what purpose and with which parameters? I am just curious because the description of the model is not referenced.
 - l. 97: “rapidly” is a relative term. Do the authors have something specific in mind that is comparatively slow?
 - l. 110 “Nonetheless,...”: I might miss something here, how exactly do the authors come up with the numbers 8% and 90%?
 - l. 113: Considering the broad level at which this analysis was performed, the authors might consider changing “strongly suggest” to “suggest”.
- Throughout the manuscript, the term “competitor” appears. In l. 86ff, “competing strains” are defined in regard to the model but the term “competitor” is used before (e.g. l. 46) and after (e.g. l.118). It might be good to check that these all refer to the same definitions, are “competitors” the same as “competing strains” and are the two strains in the experiment “competitors” just because they kill each other?
- figure 2b, I do not see the brown line that is referred to in the figure legend
- figure 2e) and f) it is not clear to me why the two panels are plotted differently. Couldn't panel e) be plotted like f) or f) analyzed over time and plotted like e)?
- figure 3, maybe I am missing something, it is not clear to me what the colours refer to in panel a-d.
- figure 3: are panel 3a and panel 3b referenced in the text?

- figure 4: is figure 4f referenced in the text?
- figure 4: does the number of T6SSs and the number of T6SS effectors correlate with genome size?
- Method section page 1: What is the rationale for 5% of the cells activating their T6SS in the model? How sensitive is the model to the number of cells that activate their T6SS?

Reviewer #2 (Remarks to the Author):

Comments on "Killing by Type VI secretion drives clonal phase separation ... " by Luke et al.

This is a very interesting MS on the topic of spatial self-organization patterns. It suggests that a novel ecological/biology (was called targeted killing of non-kin competitors in this MS) process drives the pattern formation. Their experiments with *Vibrio cholera* and theory models reveal the underlying mechanism that cause this spatial phase separation, and shown their convinced dynamics features. This is a remarkable evidence of phase separation in ecology and biology. However, there are still few point need to be clarified from my opinions. I am very happy to recommend it to be accepted after the authors deal with the following comments.

Comments:

- The introduction is too short for the reader to capture the background about self-organization pattern in biology and ecology. It seems that the current version is only relate to relevance the microbes. I would suggest include a paragraph on phase separation in active matter. Some classic references such as Nguyen et al, Emergent collective phenomena in a mixture of hard shapes through active rotation *Phys. Rev. Lett.* (2014); Liu et al, Phase separation driven by density-dependent movement: A novel mechanism for ecological patterns. *Physics of Life Reviews* (2016); Michael & Julien, Motility-induced phase separation, *Motility-Induced Phase Separation* (2015).
- L64, although the authors provide a clear define and calculate formula for the structure factor $S(q)$. This character also play key role the whole MS both in experiment and theoretical simulation (figure 2). However, I found this curves doesn't match the previous literature, for example see figure 4 of refer [37]. Hence, I would like suggest provide the computer code as supplementary materials. Does the peak of the q_{max} fall in the coarsening law? I guess this will works well from the support movies.
- L115, This is an excellent paragraphs about phase separation and Allen-Cahn equation. However, distinguishing phase separation and Turing patterns were detail discussion in recent review work (see Phase separation driven by density-dependent movement: A novel mechanism for ecological patterns. *Physics of Life Reviews* 2016). Does the present phase separation follow the coarsening law?

Reviewer #3 (Remarks to the Author):

From three categories of data, McNally et al. promote a far-reaching hypothesis, namely that evolutionary increases in the number of T6SSs carried by bacterial lineages caused subsequent increases in the proportions of those lineages' genomes dedicated to the secretion of products that benefit, rather than antagonise, diverse neighbouring competitors.

Arguing in support of this hypothesis, the authors show that a) two strains of the same bacterial species that each possess a distinct T6SS (and can therefore kill each other) grow into more highly structured patches after thorough mixing prior to growth than do mutants of those same strains lacking functional T6SSs, b) mathematical models can be constructed that predict that spatial structure generated by T6SS-mediated killing can, given certain assumptions, promote the stability of energetically costly secretions that benefit neighbours and c) across diverse taxa, the number of T6SSs present in a genome correlates significantly with the proportion of genes predicted to be associated with extracellular secretions (of any sort). The authors suggest that the first two results above indicate that the third result (the observed correlation) is causally driven by the number of T6SSs rather than by any other (unnamed) alternative.

To summarise my take on this paper, each of the data sets – experimental, theoretical and correlative – represents very nice and interesting work, but nonetheless they collectively fall short of uniquely supporting the general hypothesis advocated in this paper. The modelling is impressive and the documented correlation is novel and intriguing. Nonetheless, the experimental and modelling results are presented as being uniquely predictive of the observed correlation, but that is problematic because alternative scenarios that might generate the same prediction are not addressed (see below).

From a scholarly perspective, the paper fails to place itself in the context of previous work that has similarly proposed that antagonistic secretions (of which T6SSs represent just one subcategory) should increase spatial structure between microbes, increase local relatedness and thereby potentially promote cooperation. Among others, Strassmann et al. 2011 (*Annu Rev Microbiol*) and Lyons et al. 2016 (*Curr Biol*) have explicitly stated this hypothesis. While the experiments and modelling of the current study probe this hypothesis more deeply than previous works, the paper almost reads as though the basic idea is being proposed for the first time. That should be corrected.

Regarding the paper's main hypothesis, the authors write: "alternative hypotheses cannot be ruled out entirely". But such alternatives were not even presented, much less seriously considered. There are plausible alternative explanations of the observed correlation that the presented experimental and modelling data do not speak strongly against. For example, increases in the number of T6SSs may increase sampling of secretome genes from killed neighbours by HGT and in the long run drive increases in secretome proportions due to increased opportunity for HGT acquisitions.

Additional considerations:

1. The proposal under question is that acquisition of increasing numbers of T6SSs drives the

acquisition of an increasing proportion of the genome dedicated to cooperative secretions specifically because, absent those T6SSs, non-producing neighbours (including many different non-producing species in a community) would benefit from the secretions without cost, thereby outcompeting producers and driving the production trait to extinction.

This hypothesis seems to assume that most secretome products are beneficial to neighbours of a different T6SS type. But is there any evidence that this is the case? Many secretome products other than T6SS products may in fact be antagonistic rather than beneficial to non-kin. The proportion of the genome represented by antagonistic secretions might correlate with the number of T6SSs just as tightly as do cooperative secretions. In this case, T6SSs would not be preferentially driving increased investment in other-beneficial secretions (if T6SSs causally drive the correlation at all, which is itself not clear), but rather would be generically driving an increase in all secretions, both other-antagonistic and other-beneficial. To address this would require extensive knowledge of the fitness effects of various secretions across a wide range of taxa, a rather tall order.

Moreover, some non-antagonistic secretions that are beneficial to cells of the producer genotype may not be equally beneficial, or beneficial at all, to many of the highly divergent taxa that compose a complex microbial community.

The authors write: "these results strongly suggest that T6SS-mediated killing creates conditions that favour exoproduct evolution..." But that is different from saying that T6SS-mediated killing creates conditions that favour specifically other-beneficial exoproduct evolution more than other-antagonistic exoproduct evolution, which is what seems to be the real hypothesis the authors seek to promote.

2. Also, only the potential for secretions to benefit non-producing cells of other T6SS types seems to be considered here. The basic hypothesis requires a long-term statistical association such that the non-producers of a secreted product that could benefit from that product ("cheats") and that represent the greatest threat to its evolutionary maintenance are of different T6SS types than the producer. But non-producers of the same T6SS type (that are thus not killed by the relevant T6SS) might be as much or more destabilizing of other-beneficial secretions than non-producers of different T6SS types. Within-T6SS-type cheats need to be considered.

3. To state the obvious, bacterial species differ in many ways other than the number of T6SSs they carry. The reported experiments involved two strains of the same species that are relatively similar compared to most members of specific natural communities. Some additional important unresolved issues include how much spatial structure within microbial communities is generated by differences in the extracellular biochemistry of constituent species other than differences in T6SSs and how much T6SSs augment such structure generated by other factors.

To conclude, the authors have done some excellent and unique modelling showing that T6SSs have the potential to promote cooperation and have also uncovered an intriguing correlation that calls for explanation. But they attempt to draw (or strongly suggest) a

major causal inference that is not sufficiently supported.

Reviewer #4 (Remarks to the Author):

McNally et al. have modelled and explored experimentally contact-mediated killing in a dense bacterial system. The basic idea is that a bacterium can emit a chemical that is toxic to bacteria that are not related to it. The death of a neighbour competitor allows the bacterium to replicate. The resulting dynamics is shown to exhibit Model A coarsening behavior. The authors verified the model expectations with experiments on *Vibrio cholerae*.

This paper is acceptable for publication after some minor modifications. I think it is quite interesting, but the main interest is the experimental realisation. I did not find the result very surprising, because the killing of a neighbouring competitor in order to create free space for one's own offspring is manifestly a non-conserved ordering process in the Ising universality class. This system is mathematically analogous to Glauber dynamics in the spin-1/2 Ising model, because if the neighbour is not a kin member, then the dynamics will replace it by a kin member. This is like a spin up site flipping a neighbouring spin down site to lower the energy, resulting in a phase ordering dynamics that does not conserve total spin. It is nice that the result was seen in the experiments, but I do not know why the scaling behavior was simulated using three different techniques (individual-level, PDE and Ising), when the universality of phase ordering processes of this type was established nearly thirty years ago. One would have been enough!

The scaling behavior is presented in a confusing way. With a non-conserved order parameter, coarsening proceeds by the elimination of small domains, leaving only larger domains. These too then shrink away, leaving still larger domains. So the characteristic domain size grows with time. However, there is no peak in the structure function, because of the lack of conservation (which would force $S(q)$ to vanish at $q \rightarrow 0$). The authors plotted $qS(q)$ rather than q , which does have a peak of course but it is confusing. The alert reader will expect to see a $S(q)$ that does not vanish at the origin. I recommend that they plot this in a conventional way.

The section of the paper on the phylogenetic approach to the phase separation question was not clear to me. Perhaps I am not sufficiently familiar with the biology but I did not know what a Bayesian phylogenetic mixed model was. I also did not understand at first reading what was the hypothesis that they were testing, and how phylogeny related to experiments. I think this needs to be explained more clearly to an interdisciplinary reader, because the main impact of the paper might be this part. However, this part does not of course relate to the coarsening mechanism or the scaling laws. A biologist without any knowledge of the coarsening/scaling could have done this part. So in the end, I was confused about the real connection between this part of the paper and the scaling/coarsening.

In the conclusion of the paper, the findings are presented as being a new paradigm, akin in importance to the Turing mechanism for pattern formation. I recommend that this be toned

down. Universal non-conserved scaling and pattern formation is well-understood and in this particular context, rather obvious (but still interesting and probably novel). Whether this is a new class of active matter is really not clear to me at all, and I do not see the need for this type of hyperbole. Non-conserved domain growth is what this paper is all about, and the nice results stand on their own.

Response to referees.

Questions/comments from Referee 1 (related comments may be grouped together):

Question 1:

(title (1)) "clonal phase separation". I would be careful with the term "clonal" here. Experimental data from the T6SS field indicates that killing or no killing (immunity) only depends on the effector- and immunity protein, and is, roughly speaking, independent of the rest of the genome. In other words, two strains that are not clonal but share the same effector and immunity pairs would be immune and not separate. Although "clonal" is correct based on the way the experiments are set up in this manuscript, I recommend to at least remove "clonal" from the title and maybe somewhere in the manuscript refer to the concept that all that matters is the set of immunity genes/proteins independent of actual clonality over the whole genome.

- (title (2)) "drives ... the evolution of cooperation". Based on what is presented in this manuscript I think it is fair to conclude that T6SS killing drives phase separation but I think it is too much to conclude that T6SS-mediated killing drives the evolution of cooperation. Overall, the manuscript should have an emphasis on the results of the first half that are validated through mathematical models and experiments.

Response:

We thank the referee for these helpful comments. We agree and have replaced "clonal phase separation" with "genetic phase separation" and changed the wording in the title to "correlates with increased cooperation."

Text Changes:

Title changed to: "Killing by Type VI secretion drives genetic phase separation and correlates with increased cooperation"

Question 2:

Overall, the manuscript is very short and condensed. Considering that the results might be of interest to a wide readership, I think it would benefit from a more detailed introduction, explanation of the rationale for choosing particular models and more detailed conclusion. For example, if one of the main findings of the paper is the description of the dynamics by which the phase separation happens, how does this dynamic differs from the other forms of phase separation (discussion) and why is phase separation within biological systems itself not introduced in the introduction with more details?

Response:

We agree with the referee, and have expanded both the introduction and discussion sections of the paper to give further details on the relation of our results to other work and to make the paper more accessible to a wide readership.

Text Changes:

We have added new paragraphs on: phase separation in active matter, phase separation due to antagonism, the biological mechanisms of the T6SS, alternative hypotheses that may be consistent with our results, and 'Model A' coarsening.

Question 3:

T6SS-mediated killing among V. cholerae is used as an experimental model system in this manuscript. Experimental work has been performed on the T6SS of V. cholerae (Pukatzki et al., 2006, Basler et al., 2012), contact-dependent prokaryotic killing by V. cholerae via the T6SS (MacIntyre et al., 2010) and killing between V. cholerae strains based on the presence/absence of effector- and immunity genes (Dong et al. 2013, Unterweger et al., 2014) and should be cited. A study not using V. cholerae but Burkholderia thailandensis indicates that the T6SS might be used to get rid of cheats that do not contribute to quorum-sensing related costs (Majerczyk et al., 2016). This paper might be of interest to the authors and I think should be referred to in the context of T6SS and cooperative traits. Although the paper by Borenstein et al. is referenced, I recommend putting the results and the models used here a little more in the context of the results and models of Borenstein et al.

Response:

These are excellent references, thank you for bringing them to our attention. We have added citations to all of the above papers (as well as others that are relevant) and discuss their relation to our results in the expanded introduction and discussion.

Text Changes:

We added the suggested references and discuss how they relate to our results in the main text:

Here we use these citations, among others, to discuss T6SS background (from the intro).

The T6SS is a potent mechanism of bacterial aggression that can deliver effector proteins directly into eukaryotic cells to mediate virulence by cellular disruption, and into adjacent bacteria to mediate competition by killing non-kin while leaving kin with corresponding protective immunity proteins unscathed^{23,24}. In *Vibrio cholerae*, T6-proficient strains utilize the T6SS to intoxicate T6-deficient eukaryotic predators and diverse proteobacteria, as well as other more closely related *V. cholerae* isolates that lack identical effector immunity pairs²⁵⁻³¹. T6-mediated segregation occurs during co-culture of T6-proficient *V. cholerae* with T6-deficient *E. coli*. Segregation was also predicted to occur between two mutually antagonistic T6-proficient strains³², and recently demonstrated at the single cell level in co-cultures of *V. cholerae* and *Aeromonas hydrophila*²⁰.

Here we discuss how the above papers relate to our work (from the results):

Our models and experiments demonstrate that T6SS-mediated killing can generate favourable conditions for the evolution of public-goods cooperation^{1,12,15,44}. This can occur in two ways. Firstly, T6SS-mediated killing induces positive frequency-dependent selection, allowing cooperators to resist rare cheats. Secondly, T6SS-mediated killing precipitates self-organized structuring of microbial communities, reducing the diversity of cheats that can exploit cooperators. In addition to the predictions of our models previous experimental work has suggested that, via pleiotropic linkage to quorum sensing communication

systems, the T6SS can also act as a policing mechanism protecting against the evolution of quorum sensing cheats⁴⁵. Together this suggests that there are three complementary paths via which T6SS-mediated killing could favor the evolution of cooperation.

Question 4:

Figure 2b shows how sensitive the phase separation described in here is to the starting ratios at which the two strains are mixed in an experiment. This should be mentioned and commented on in the text. Did the authors determine the actual ratio of the two strains at some point by plating out the mix and counting CFUs? Do I read this correctly in Figure S1 that the bacteria have been mixed at a 1:10 ratio in this experiment?

Response:

This is an excellent question, as it is vitally important that this is clear in the manuscript. If the starting ratios of the two strains are too far off, one “wins” within the time period of experimental observation. But, as Figure 2C shows, even populations starting at different ratios generate structure with the same $t^{1/2}$ dynamics. This is one of the benefits of a transition that exhibits universality, as ‘Model A’ does; many of the microscopic details do not affect the coarsening process.

Text Changes: Text was added to our discussion of Figure 2c to highlight that samples – even with different starting ratios – behaved similarly:

To demonstrate this equivalence across wavenumbers, we plot $q_m S(q)$ versus q/q_m (Fig. 2d). This collapses all data onto one master curve. In fact, due to the universality of non-conserved domain growth, this collapse could have been expected. Importantly, this universality, shown in Fig. 2c and d, demonstrates that while initial conditions – like the initial number ratio of the two competing strains – may influence the timing of phase separation, they do not influence how phase separation occurs, or that the characteristic clonal group size always grows as \sqrt{t} .

Minor Questions and comments:

We deeply appreciate the referee’s incredible attention to the details our paper. This certainly took a lot of time, and it goes above and beyond the normal call of referee duty. Thank you! Below, we address their many minor comments.

- l. 23 “novel” not necessary

We have removed the word novel.

- l. 28f I can see from the data presented later on how mathematic models predict that T6SS-mediated killing created spatial patterns that potentially favour cooperative traits. It is not clear to me, how the mathematical model makes the prediction about the “evolution” of such traits as referred to in this sentence. I also think that the results of the analysis shown in the last figure do not actually “verify” this prediction as phrased here.

The model presented in Figure 3 does show that T6SS-mediated killing should favour the evolution of cooperation by showing that this killing allows cooperative strains to win in competition with cheaters where they would otherwise be outcompeted. We have clarified

this in the results section of the manuscript. We have changed the word “verify” to “support”.

- l. 31 based on what is presented in the manuscript, the final sentence of the abstract might rather read “...while simultaneously creating conditions potentially favoring cooperation with kin”.

We have made this change.

- l. 40ff “No mechanisms have been described...”: I consider this a very bold statement. Maybe good to specify what “mechanism” refers to here. If “mechanism” refers to the biological mechanism of the T6SS, T6SS-mediated killing has been described in the past (see references mentioned above). What about formulating this sentence as a question, something like “Considering the ecological relevance of dense, well-mixed communities displaying no net growth, we were wondering if microbes are able to genetically segregate in such communities.”

We have modified our introduction removing this sentence.

- l. 45ff, this sentence could be potentially misleading. Effectors are delivered to neighbouring cells independently of what they are. Maybe modification as follows in l. 46f would help “... proteins to adjacent cells, competitors die whereas clonemates with ...)

We have modified this sentence.

- l.47, re-consider references for this sentence. Publications for prokaryotic killing of *V. cholerae* and protection by effector-immunity interactions that could be cited here are: MacIntyre et al., 2010; Dong et al., 2013, Brooks et al., 2013. In more general terms of antiprokaryotic killing via the T6SS independent of the species, also the following papers could be cited: Hood et al., 2010, Russell et al., 2011 or reviews such as Ho et al., 2014; Cianfanelli et al., 2016

We have changed the references for this sentence and expanded our introductory section on T6 and *Vibrio cholerae*.

- l. 47, l.27 “new class of active matter”: the wording sounds a little too dramatic to me.

We have changed this sentence.

Text changed to:

Physically, this system bears similarities to active matter^{18,48,52,53}; phase separation has also been observed in these far from equilibrium active systems, wherein constituents expend energy to move.

- l. 54 rationale for why these three models were chosen would be helpful.

We have added rationale for the choice of models. Specifically, we now say:

IBMs are appealing, in that they offer an intuitive simulation of discretized, interacting individuals. However, IBMs often lack mathematical transparency, limiting generalization. We thus modeled our system using two distinct, mathematically-defined approaches: an ecologically-based Partial Differential Equation model in order to gain analytical insight into the dynamics (PDE; Fig. 1c; Supplementary Movie 2), and the classic Ising spin model

in order to relate our results to classical modeling of phase separation in statistical mechanics³⁴ (Fig. 1d; Supplementary Movie 3).

- l. 62ff the variable q (wavelength) is described in the figure legend but not the next. The authors might consider adding a description of q to the text.

We have added a description of q in the text.

- l. 75 for a reader not familiar with the Model A phase separation process, it might be helpful providing more background, for example on other processes that are described with this equation

We have added an explanation and example.

- l. 82 it is not clear why the IBM was chosen in Figure 2 among the three models used in Figure 1

We simply chose one model to illustrate the patterns, without overloading the reader with plots. The other two models illustrated in Supplementary Figure 2.

- l. 82 ff "T6SS-mediated killing...": The authors should consider disconnecting the experimental results in this paper from the results in the referenced papers by replacing the comma with a period. Based on the findings of the referenced papers, it is likely that the longer length scales protect diffusible public goods from cheats here but it has not been tested in this exact setting studied here.

We have made this change.

- l. 86: No rationale was mentioned why the PDE model is used here out of the three models that were used at the beginning.

We have added a sentence explaining that a PDE approach was used in order to allow us to derive analytical as well as numerical results.

- l. 86ff: Has such a model of a cooperator and a non-producing cheat been used before in a different context and if so for what purpose and with which parameters? I am just curious because the description of the model is not referenced.

While there have been many models of cooperation and cheating we don't know of any using the same model structure as used here.

- l. 97: "rapidly" is a relative term. Do the authors have something specific in mind that is comparatively slow?

The referee is correct. We have removed the word rapidly.

- l. 110 "Nonetheless,...": I might miss something here, how exactly do the authors come up with the numbers 8% and 90%?

We have added description of how these values are calculated in the methods section.

- l. 113: Considering the broad level at which this analysis was performed, the authors might consider changing "strongly suggest" to "suggest".

We have made this change.

- Throughout the manuscript, the term “competitor” appears. In l. 86ff, “competing strains” are defined in regard to the model but the term “competitor” is used before (e.g. l. 46) and after (e.g. l.118). It might be good to check that these all refer to the same definitions, are “competitors” the same as “competing strains” and are the two strains in the experiment “competitors” just because they kill each other?

In all cases “competitors” refer to competing strains and has the same meaning.

- figure 2b, I do not see the brown line that is referred to in the figure legend

This brown line is obscured by the purple line in the graph which shows almost identical values. We now note this in the legend.

- figure 2e) and f) it is not clear to me why the two panels are plotted differently. Couldn't panel e) be plotted like f) or f) analyzed over time and plotted like e)?

The difference in x-axis reflects a key difference in between simulations, where we have lots of different timepoints, and experiments, where everything is harvested at the same time. In the simulations (Figure 2e), we analyze the growth of population structure over time, while in the experiments (Figure 2f), we analyze the *degree* of structure across different length scales.

- figure 3, maybe I am missing something, it is not clear to me what the colours refer to in panel a-d.

The colours are simply to help distinguish the lines for different starting frequencies. This has been noted in the figure legend.

- figure 3: are panel 3a and panel 3b referenced in the text?

We now reference these in the text.

- figure 4: is figure 4f referenced in the text?

We now reference this in the text.

- figure 4: does the number of T6SSs and the number of T6SS effectors correlate with genome size?

They do show weak correlations with genome size, as illustrated in Supplementary Figure 6. However, this correlation is not the driver of our results as a model including genome size as a predictor still showed significant effects of the number of T6SSs and T6SS effectors on secretomes size (Supplementary Table 5).

- Method section page 1: What is the rationale for 5% of the cells activating their T6SS in the model? How sensitive is the model to the number of cells that activate their T6SS?

This is a good point. These values simply dictate how finely-grained the temporal dynamics of the simulation are: allowing fewer cells to kill and reproduce creates less change per update. We now explain this in the Methods section: “Rates of killing and reproduction were chosen to provide sufficient temporal resolution of population dynamics while still being computationally efficient.”

Questions/comments from Referee 2 (related comments may be grouped together):

Question 1:

The introduction is too short for the reader to capture the background about self-organization pattern in biology and ecology. It seems that the current version is only relate to relevance the microbes. I would suggest include a paragraph on phase separation in active matter. Some classic references such as Nguyen et al, Emergent collective phenomena in a mixture of hard shapes through active rotation Phys. Rev. Lett. (2014); Liu et al, Phase separation driven by density-dependent movement: A novel mechanism for ecological patterns. Physics of Life Reviews (2016); Michael & Julien, Motility-induced phase separation, Motility-Induced Phase Separation (2015).

Response:

Thank you for this comment. We certainly agree with the referee that the manuscript should include more information about self-organization in biology. We spent a lot of time trying to work this into the introduction, but it never worked well with the flow of the MS. In hindsight, the reason for this was clear: the phase separation / physical basis of pattern formation is a *result* of our work, but it was not an effective motivator for the work. Put another way, we needed to describe our results showing that phase separation occurred before we could talk about phase separation in biology in general. We now include a more extensive discussion of this topic, but place it in the discussion rather than the introduction.

Text Changes:

A new section on active matter was added:

Physically, this system bears similarities to active matter^{18,50,55,56}; phase separation has also been observed in these far from equilibrium active systems, wherein constituents expend energy to move. Phase separation in these systems typically occurs due to differences in mobility as a function of density; constituents move slowly through crowded regions, and quickly through low density regions. Mobility-induced phase separation has been observed (or predicted) in systems as varied as swimming bacteria⁵⁹, self-propelled colloids^{60,61}, mussels⁵⁰, granular rods⁶², active filaments^{63,64}, rotating particles^{65,66}, among other systems⁵¹. In the current system, activity is derived from reproduction and killing events at high density rather than constituent mobility⁶⁷, leading to a 'Model A' transition.

Question 2:

L64, although the authors provide a clear define and calculate formula for the structure factor $S(q)$. This character also play key role the whole MS both in experiment and theoretical simulation (figure 2). However, I found this curves doesn't match the previous literature, for example see figure 4 of refer [37]. Hence, I would like suggest provide the computer code as supplementary materials. Does the peak of the q_{max} fall in the coarsening law? I guess this will works well from the support movies.

Response:

The referee is absolutely correct. After extensive internal debate, we originally decided to normalize $S(q)$ by q to make the curves (and the characteristic q 's of each population) more

intuitive to biologists, but of course this is confusing to physicists that are comfortable with the traditional ways of doing things. In light of your comments, as well as those of Referee 4, we have decided to plot these curves in the traditional manner.

Text Changes: We replotted Figure 2 the traditional way, and included an explanation in the text.

Question 3:

L115, This is an excellent paragraphs about phase separation and Allen-Cahn equation. However, distinguishing phase separation and Turing patterns were detail discussion in recent review work (see Phase separation driven by density-dependent movement: A novel mechanism for ecological patterns. Physics of Life Reviews 2016). Does the present phase separation follow the coarsening law?

Response:

We also agree that Turing patterns should be addressed directly. While slow maturation of fluorescent proteins makes it difficult to generate experimental timelapses over wide ranges of times, we utilize the fact that during coarsening the structure factor, $S(q)$, and the mean wavevector, q_m , change over time. Thus, the Allen-Cahn coarsening law, $q_m \sim 1/\sqrt{t}$, which is different than a Turing pattern ($q_m \sim \text{constant}$) or Cahn-Hilliard coarsening ($q_m \sim 1/t^{1/3}$), becomes $S(q_m) \sim 1/q_m$. We test this prediction by plotting $S(q_m)$ vs. q_m and $S(q) \cdot q_m$ vs. q/q_m . We now calculate $S(q)$ the traditional (and proper) way, making agreement with coarsening predictions shown in Figs. 2c and d more straight forward.

Text Changes: A section was added discussing these concepts:

Phase separation is well-known to drive pattern formation in biology^{48,49}, but has mainly been investigated using either Turing activator-inhibitor feedbacks^{50,51}, or positive density-dependent movement, described by the Cahn-Hilliard equation^{48,52-54}. In this paper we describe a third general mechanism of self-organized pattern formation: targeted killing of non-kin competitors. This drives a 'Model A' phase separation; the kinetics of this coarsening process—described by the Allen-Cahn equation—only depend on a few cellular details. While we explore this process in bacteria, it is probably more general, applying to other organisms that kill adjacent non-kin (*e.g.*, allelopathy in plants⁵⁵ and animals⁵⁶). However, while 'Model A' coarsening is universal, the realization of such dynamics in a densely packed, immobile, athermal system is likely unique to biology.

Questions/comments from Referee 3 (related comments may be grouped together):

Question 1:

From a scholarly perspective, the paper fails to place itself in the context of previous work that has similarly proposed that antagonistic secretions (of which T6SSs represent just one subcategory) should increase spatial structure between microbes, increase local relatedness and thereby potentially promote cooperation. Among others, Strassmann et al. 2011 (Annu Rev Microbiol) and Lyons et al. 2016 (Curr Biol) have explicitly stated this hypothesis. While the experiments and modelling of the current study probe this hypothesis more deeply than

previous works, the paper almost reads as though the basic idea is being proposed for the first time. That should be corrected.

Response:

This is an excellent suggestion. We agree that we gave prior literature less attention than was ideal. This was largely due to the limitations on the prior (*Nature*) submission format. We now have expanded the discussion of antagonistic interactions in the role of generating spatial structure, including the citations suggested above (as well as others).

Text Changes: We have added the following paragraphs to the introduction:

It is widely recognized that the spatial segregation of cooperative microbes away from cheats can solve this cooperative dilemma by ensuring that the investment of cooperators goes to other adjacent cooperative individuals^{1,5,10-12}. Mechanisms creating assortment when organisms expand their ranges via growth into free space have recently received much attention¹³⁻¹⁸, where robust patterns of genetic segregation can occur via stochastic bottlenecks. However this mechanism cannot generate genetic segregation within dense, well-mixed communities displaying no net growth, despite the clear ecological relevance of such communities.

One mechanism that has been proposed to potentially generate spatial structure in dense communities is antagonistic interactions among genotypes^{1,19-22}. If different genotypes interact antagonistically then wherever a genotype is in the minority they will be killed by competitors at a high rate, resulting in genetically homogenous patches. While mechanisms via which individuals can recognize and kill non-kin have been extensively studied, the consequences of such interactions for the spatial structure of communities have not been explored in detail.

Question 2:

Regarding the paper's main hypothesis, the authors write: "alternative hypotheses cannot be ruled out entirely". But such alternatives were not even presented, much less seriously considered. There are plausible alternative explanations of the observed correlation that the presented experimental and modelling data do not speak strongly against. For example, increases in the number of T6SSs may increase sampling of secretome genes from killed neighbours by HGT and in the long run drive increases in secretome proportions due to increased opportunity for HGT acquisitions.

Response:

We now consider multiple hypotheses for why strains with greater numbers of T6SSs dedicate greater proportions of their genomes to secretions. Specifically these are:

- 1) That the T6SS drives phase separation favouring cooperation.
- 2) That by inducing positive frequency dependent dynamics the T6SS can favour cooperation via "hitchhiking" (as illustrated in figure 3c).
- 3) That via linkage to quorum sensing systems the T6SS can act as a policing mechanism.
- 4) That the T6SS increases sampling of genes from killed neighbours via HGT, increasing secretome size.

5) That some uncontrolled ecological factor independently favours both T6SS and extracellular secretions.

We see hypotheses 1-3 as complementary, while hypotheses 4 and 5 invoke different forms of explanation. All hypotheses are non-exclusive however. Indeed, the work in this MS shows that in a spatial environment the effects in both 1 and 2 necessarily co-occur and combine to favour cooperation. We believe that hypothesis 4 is the least likely of all, as we would expect that genes coding for secretions and those that do not would be sampled via HGT at the same rate. While for mobile genetic elements such as plasmids an increased rate of HGT would be expected to bias gene content towards secretions, the same logic does not necessarily apply to integrated genes in this context. We now discuss these 5 hypotheses in the revised paper, and point out that further evidence is necessary to ascertain their relative roles in explaining the links between T6SS and cooperation.

Text Changes: Two paragraphs have been added weighing up alternative hypotheses for the observed correlation:

Our models and experiments demonstrate that T6SS-mediated killing can generate favourable conditions for the evolution of public-goods cooperation^{1,12,15,44}. This can occur in two ways. Firstly, T6SS-mediated killing induces positive frequency-dependent selection, allowing cooperators to resist rare cheats. Secondly, T6SS-mediated killing precipitates self-organized structuring of microbial communities, reducing the diversity of cheats that can exploit cooperators. In addition to the predictions of our models previous experimental work has suggested that, via pleiotropic linkage to quorum sensing communication systems, the T6SS can also act as a policing mechanism protecting against the evolution of quorum sensing cheats⁴⁵. Together this suggests that there are three complementary paths via which T6SS-mediated killing could favor the evolution of cooperation.

.....

While the above analysis is consistent with the predictions of our mathematical model showing that phase separation should favor the evolution of cooperation (Fig. 3), it is not conclusive. As with any broad-scale phylogenetic analysis, alternative mechanisms explaining this correlation cannot be ruled out. For example, some unknown aspect of bacterial ecology may independently select for both investment in T6SS-mediated killing and exoproduct production. By controlling for phylogenetic variation, our analysis should capture some of this of this ecological variance, through it does not eliminate it entirely. Alternatively, increases in the presence of dead competitors owing to T6SS-mediated killing can create additional opportunities for horizontal gene transfer⁴⁸, which could allow for increased acquisition of genes coding for secretions. However, such an explanation relies on horizontal gene transfer being biased towards genes encoding extracellular secretions. While there is evidence that this bias towards secretions is the case for plasmids and other mobile genetic elements⁴⁹, whether this bias occurs when sampling the genes of dead competitors remains to be determined. Detailed analysis of the phylogenetic dynamics of T6SS genes and secretions in individual clades should allow for the disentanglement of the relative contributions of these hypotheses driving our observed correlation.

Question 3:

The proposal under question is that acquisition of increasing numbers of T6SSs drives the acquisition of an increasing proportion of the genome dedicated to cooperative secretions specifically because, absent those T6SSs, non-producing neighbours (including many different non-producing species in a community) would benefit from the secretions without cost, thereby outcompeting producers and driving the production trait to extinction.

This hypothesis seems to assume that most secretome products are beneficial to neighbours of a different T6SS type. But is there any evidence that this is the case? Many secretome products other than T6SS products may in fact be antagonistic rather than beneficial to non-kin. The proportion of the genome represented by antagonistic secretions might correlate with the number of T6SSs just as tightly as do cooperative secretions. In this case, T6SSs would not be preferentially driving increased investment in other-beneficial secretions (if T6SSs causally drive the correlation at all, which is itself not clear), but rather would be generically driving an increase in all secretions, both other-antagonistic and other-beneficial. To address this would require extensive knowledge of the fitness effects of various secretions across a wide range of taxa, a rather tall order.

Moreover, some non-antagonistic secretions that are beneficial to cells of the producer genotype may not be equally beneficial, or beneficial at all, to many of the highly divergent taxa that compose a complex microbial community.

The authors write: “these results strongly suggest that T6SS-mediated killing creates conditions that favour exoproduct evolution...” But that is different from saying that T6SS-mediated killing creates conditions that favour specifically other-beneficial exoproduct evolution more than other-antagonistic exoproduct evolution, which is what seems to be the real hypothesis the authors seek to promote.

Response:

Thank you for the highlighting a fundamental concept that was not clear in the original manuscript. Production of costly exoproducts, whether beneficial or antagonistic to non-clonemates, can be thought of as ‘cooperative’ traits (among producer cells) that may be subject to social exploitation, because they are costly to produce but provide benefits (e.g., reduced competition) to resistant cells. Indeed, cheats readily evolve to free-ride on the antibiotics produced by others. We have clarified that even antagonistic secretions can be considered cooperative from the producer genotype’s perspective.

Text Changes: The following text was added:

As a first-order proxy for cooperativity, we measured the proportion of a clone’s genome dedicated to extracellular secretions (henceforth referred to as ‘secretome size’). While many of these secretions may have antagonistic effects on other microbes, they can still be seen as cooperative from the producing cell’s perspective, as their kin can benefit from the reduced competition that they create^{46,47}.

Question 4:

Also, only the potential for secretions to benefit non-producing cells of other T6SS types seems to be considered here. The basic hypothesis requires a long-term statistical association such that the non-producers of a secreted product that could benefit from that product (“cheats”) and that represent the greatest threat to its evolutionary maintenance are of different T6SS types than the producer. But non-producers of the same T6SS type (that are thus not killed by the relevant T6SS) might be as much or more destabilizing of other-beneficial secretions than non-producers of different T6SS types. Within-T6SS-type cheats need to be considered.

Response:

It is true that non-producer cells of the same T6SS type can still cheat on producer cells, and so the T6SS does not provide protection against these cheats. Our model does consider this type of interaction, as illustrated in figure 3a and 3b. More broadly, we believe that interactions between bacterial taxa with different T6SSs are likely to be common: T6SSs are common (25% of gram negative bacteria contain one) and diverse. Our model shows that the T6SS can provide protection against these cheats of different T6SS type (figure 3c and 3d). The reason for our hypothesis that T6SS should therefore favour cooperation overall is therefore derived from the reduced *total diversity* of cheats that can potentially exploit cooperators that possess T6SS. While such cooperators will no doubt suffer from exploitation by cheats of the same T6SS, the total level of cheating they will suffer will be reduced compared to those without T6SS that could potentially be exploited by many more strains. We have clarified and expanded upon this point in the revised manuscript.

Text Changes: When discussing the results of Figure 3 we now point out that our no killing condition corresponds to cheats of the same T6SS type as the cooperator:

We find that T6SS-mediated killing protects cooperation in two different ways. In a non-spatial (*i.e.*, constantly mixed) environment, T6SS-mediated killing can allow cooperators to resist invasion by rare cheats owing to the cooperators’ numerical dominance in antagonistic interactions (*i.e.*, it creates positive frequency-dependence (Figure 3c), while without T6SS-mediated killing (either because strains lack T6SS, or because the cheat is of the same T6SS type as the cooperator) cheats outcompete cooperators at all starting frequencies (Figure 3a).

In addition, we have added text clarifying that the expected effect of T6SS in protecting cooperation occurs because of a reduction in the diversity of cheats that can exploit cooperators:

All else equal, genotypes that possess a greater number of T6SSs should form more highly-structured patches (higher r), phase separating with a greater proportion of competitor genotypes (*i.e.*, those with non-complementary effector / immunity proteins). As a result, we hypothesize that genotypes with more T6SSs should experience less pressure from social cheating imposed by distantly-related competitors. There is, of course, an important caveat to this scenario: while T6SS-based phase separation should effectively exclude competitors, it does not address the *de novo* evolution of cheating from within the clone (*e.g.*, Figs. 3a and b).

Question 5:

To state the obvious, bacterial species differ in many ways other than the number of T6SSs they carry. The reported experiments involved two strains of the same species that are relatively similar compared to most members of specific natural communities. Some additional important unresolved issues include how much spatial structure within microbial communities is generated by differences in the extracellular biochemistry of constituent species other than differences in T6SSs and how much T6SSs augment such structure generated by other factors.

Response:

We completely agree! Many factors may affect the degree of spatial structure generated in natural communities, and these are certainly meritorious of further study. We chose to focus on among-*Vibrio cholerae* competition for two reasons: First, this scale of comparison is most important for understanding the evolution of social behavior. From an inclusive fitness perspective, there is an important difference between cooperating or competing with another member of the same species in the same population and an unrelated competitor species. The outcome of the interaction may affect the indirect fitness of the actor in the former scenario, but not the latter. Second, comparisons among *V. cholerae* strains (including isogenic non-killing controls) minimize potentially confounding differences between competitors, allowing us to identify the role played by the T6SS with maximal clarity.

Text Changes:

We have modified our concluding paragraph to stress that T6SS is obviously not the only factor at play in shaping community structure, but that our results suggest it is an important one (relevant text bolded for clarity below):

In recent years, there has been a growing appreciation that many microbial behaviors requiring extracellular metabolism are susceptible to social exploitation. Here we show how simple cell-cell aggression can, as a consequence, create a structured population favourable to cooperation. **Clearly, many factors contribute to the structure and function of microbial communities**^{1,4,15,16,19-22,41,46}. However, because T6SSs are common (found in ~25% of gram negative bacteria⁶⁸), and microbes often live in dense communities, phase-separation driven by contact-mediated killing may play a fundamental role in defining the genetic composition and ecosystem-level functionality of microbial communities worldwide.

Questions/comments from Referee 4 (related comments may be grouped together):

Question 1:

The main interest is the experimental realisation. I did not find the result very surprising, because the killing of a neighbouring competitor in order to create free space for one's own offspring is manifestly a non-conserved ordering process in the Ising universality class. This system is mathematically analogous to Glauber dynamics in the spin-1/2 Ising model, because if the neighbour is not a kin member, then the dynamics will replace it by a kin member. This is like a spin up site flipping a neighbouring spin down site to lower the energy, resulting in a phase ordering dynamics that does not conserve total spin. It is nice that the result was seen

in the experiments, but I do not know why the scaling behavior was simulated using three different techniques (individual-level, PDE and Ising), when the universality of phase ordering processes of this type was established nearly thirty years ago. One would have been enough!

Response:

We agree with the referee, and include three models to emphasize the universality to a broad readership who may not be familiar with this concept. However, we have edited the manuscript to make it clear that this universality is to be expected.

Text Changes:

Text was added to emphasize the agreement with the Ising universality class:

To demonstrate this equivalence across wavenumbers, we plot $q_m S(q)$ versus q/q_m (Fig. 2d). This collapses all data onto one master curve. In fact, due to the universality of non-conserved domain growth, this collapse could have been expected. Importantly, this universality, shown in Fig. 2c and d, demonstrates that while initial conditions – like the initial number ratio of the two competing strains – may influence the timing of phase separation, they do not influence how phase separation occurs, or that the characteristic clonal group size always grows as \sqrt{t} .

Question 2:

The scaling behavior is presented in a confusing way. With a non-conserved order parameter, coarsening proceeds by the elimination of small domains, leaving only larger domains. These too then shrink away, leaving still larger domains. So the characteristic domain size grows with time. However, there is no peak in the structure function, because of the lack of conservation (which would force $S(q)$ to vanish at $q \rightarrow 0$). The authors plotted $qS(q)$ rather than $S(q)$, which does have a peak of course but it is confusing. The alert reader will expect to see a $S(q)$ that does not vanish at the origin. I recommend that they plot this in a conventional way.

Response:

The referee is absolutely correct. After extensive internal debate, we originally decided to normalize $S(q)$ by q to make the curves (and the characteristic q 's of each population) more intuitive to biologists, but of course this is confusing to physicists that are comfortable with the traditional ways of doing things. In light of your comments, as well as those of Referee 2, we have decided to plot these curves in the traditional manner.

Text Changes: We replotted Figure 2 the traditional way, and included an explanation in the text.

Question 3:

The section of the paper on the phylogenetic approach to the phase separation question was not clear to me. Perhaps I am not sufficiently familiar with the biology but I did not know what a Bayesian phylogenetic mixed model was. I also did not understand at first reading what was the hypothesis that they were testing, and how phylogeny related to experiments. I think this needs to be explained more clearly to an interdisciplinary reader, because the main

impact of the paper might be this part. However, this part does not of course relate to the coarsening mechanism or the scaling laws. A biologist without any knowledge of the coarsening/scaling could have done this part. So in the end, I was confused about the real connection between this part of the paper and the scaling/coarsening.

Response: We understand where the referee is coming from. In the extremely brief presentation of our original paper, the connection between key elements was given short shrift, which is especially confusing given the interdisciplinary nature of our work.

Text Changes: We have expanded our discussion of the phylogeny, including our motivation for the analysis, our rationale for the comparisons we conducted (specifying how it relates to our experiments and models), our hypothesis for the correlation between T6SS and secretome size we discovered, and how this result can be interpreted. Sorry for the relatively long cut and paste here, but this is the relevant section:

Does T6SS-mediated killing have a similar effect in the real world, where ephemeral resources, physical disturbance, and intense competition may impede these mechanisms? We approach this question phylogenetically, examining the relationship between the proportion of each genome coding for potentially-exploitable extracellular products and its T6SS complexity, with the rationale that microbes possessing a greater number of T6SSs may face less social exploitation by living in more structured communities. All else equal, genotypes that possess a greater number of T6SSs should form more highly-structured patches (higher r), phase separating with a greater proportion of competitor genotypes (*i.e.*, those with non-complementary effector / immunity proteins). As a result, we hypothesize that genotypes with more T6SSs should experience less pressure from social cheating imposed by distantly-related competitors. There is, of course, an important caveat to this scenario: while T6SS-based phase separation should effectively exclude competitors, it does not address the *de novo* evolution of cheating from within the clone (*e.g.*, Fig. 3a and b).

As a first-order proxy for cooperativity, we measured the proportion of a clone's genome dedicated to extracellular secretions (henceforth referred to as 'secretome size'). While many of these secretions may have antagonistic effects on other microbes, they can still be seen as cooperative from the producing cell's perspective, as their kin can benefit from the reduced competition that they create^{46,47}. We constructed a Bayesian phylogenetic mixed model of T6SS-containing Proteobacteria and Bacteroidetes (Fig. 4a, Supplementary Fig. 6) using 439 genomes from 26 genera. Secretome size is positively correlated with both the number of T6SSs (Fig. 4b, d, Supplementary Table 2) and T6SS effector proteins (Fig. 4c, e, Supplementary Table 2) present, and the model shows an excellent overall fit to the data, explaining 99% of the variance in secretome size (Fig. 4f). These results are also robust in univariate analyses (Supplementary Tables 3 and 4) and to the inclusion of genome size as a predictor (Supplementary Table 5). As our analyses include many closely related strains (*e.g.*, many *Helicobacter pylori*, Fig. 4a), most (91%) of the variance in secretome size is explained by the phylogenetic relationships among strains. Nonetheless, the number of T6 secretion systems and T6SS effectors are important predictors of secretome size, explaining 8% of the total, and 90% of the non-phylogenetic variance in secretome size.

While the above analysis is consistent with the predictions of our mathematical model showing that phase separation should favor the evolution of cooperation (Fig. 3), it is not conclusive. As with any broad-scale phylogenetic analysis, alternative mechanisms explaining this correlation cannot be ruled out. For example, some unknown aspect of bacterial ecology may independently select for both investment in T6SS-mediated killing and exoproduct production. By controlling for phylogenetic variation, our analysis should capture some of this of this ecological variance, through it does not eliminate it entirely. Alternatively, increases in the presence of dead competitors owing to T6SS-mediated killing can create additional opportunities for horizontal gene transfer⁴⁸, which could allow for increased acquisition of genes coding for secretions. However, such an explanation relies on horizontal gene transfer being biased towards genes encoding extracellular secretions. While there is evidence that this bias towards secretions is the case for plasmids and other mobile genetic elements⁴⁹, whether this bias occurs when sampling the genes of dead competitors remains to be determined. Detailed analysis of the phylogenetic dynamics of T6SS genes and secretions in individual clades may allow the relative contributions of these hypotheses to our observed correlation to be disentangled.

Question 4:

In the conclusion of the paper, the findings are presented as being a new paradigm, akin in importance to the Turing mechanism for pattern formation. I recommend that this be toned down. Universal non-conserved scaling and pattern formation is well-understood and in this particular context, rather obvious (but still interesting and probably novel). Whether this is a new class of active matter is really not clear to me at all, and I do not see the need for this type of hyperbole. Non-conserved domain growth is what this paper is all about, and the nice results stand on their own.

Response:

We thank the referee for this comment; we agree, the original statement was hyperbolic. We have toned the statement down to a level we think is more appropriate.

Text Changes:

We modified that statement to read:

Physically, this system bears similarities to active matter^{18,19,36,37}, except activity is derived from reproduction and killing events at high density rather than constituent mobility⁴⁰.

REVIEWERS' COMMENTS:

Reviewer #1 (Remarks to the Author):

The authors addressed the comments to my satisfaction.

Reviewer #2 (Remarks to the Author):

The revised version is deal with my previous questions. I am happy to recommend it published as present version.

Reviewer #3 (Remarks to the Author):

Question 1:

This new material on this theme is good. Entirely at their discretion, the authors might also consider citing Rendueles, Amherd & Velicer 2015, (Current Biology 25: 1673–1681) when they cite previous work that has proposed that microbial antagonisms might promote cooperation and when they discuss the potential for such antagonisms to generate positive frequency dependence (PFD) of fitness that in turn purges local within-patch diversity while generating inter-patch spatial structure that could promote cooperation. In that study, Rendueles et al. experimentally demonstrated the occurrence of PFD due to inter-strain antagonisms that the current authors propose hypothetically. Rendueles et al. also showed that PFD can simultaneously purge local within-patch diversity while promoting inter-patch diversity across a larger spatial scale and suggested that such spatial structure generated by PFD antagonisms (including any caused by Type VI secretion systems) could promote cooperation.

Question 2:

I find the manuscript greatly improved by the careful presentation of multiple possible explanations of the observed correlation. Regarding the fourth hypothesis, I would argue that it is more plausible than the authors suppose, because it would seem much easier to complexify a microbial cell's exobiochemistry without negative fitness effects than that of its inner workings. I would guess that a larger proportion of foreign genes taken in by transformation (or whatever means) that affect a cell's exome would be neutral or beneficial than of foreign genes encoding products that remain inside the cell. However, because the authors present the fourth hypothesis straightforwardly, I don't think any change to the text needs to be made on this point.

Question 3:

In their rebuttal, the authors write "even antagonistic secretions can be considered cooperative from the producer genotype's perspective."

Yes, this is true. But the directionality of antagonism vs. cooperation is key here. The authors are proposing that a T6SS suite in species A could promote the stability of a distinct

extracellular Secretion X by Species A because it prevents Species B from exploiting, i.e. benefiting from, Secretion X. But if Secretion X is antagonistic toward Species B rather than beneficial (like the T6SS of Species A), Species B is inherently unable to exploit Secretion X. In this scenario, the T6SS suite of Species A inherently does not protect producers of Secretion X from exploitation by Species B because the latter are harmed rather than helped by Secretion X. The only cells that could exploit antagonistic Secretion X are non-producers of X that remain resistant to X and that presumably share the same T6SS suite as producers, thus making T6SS identity irrelevant to the dynamics of cooperation and exploitation with respect to Secretion X.

To put this more succinctly, when a T6SS suite and a distinct secretion (say X) are both antagonistic toward the same target, neither protects the other from exploitation by that target. The point of my previous comment was that some secretions encoded in the same genome as a given T6SS suite might antagonize in the same direction at the T6SSs, thus eliminating them from the set of secretions that T6SSs might protect from exploitation. I think the authors should acknowledge this point. Whether this scenario applies to a substantial fraction of a given genome's exomic genes would need to be determined empirically.

Reviewer #4 (Remarks to the Author):

The authors have responded to the comments well, and I believe that the manuscript can be accepted at this point. I still feel that it was overkill to have three different types of models, which the authors say is necessary to demonstrate universality to a biology-focused audience, but I think that this could have been done in a citation to the huge literature, leaving more space for the biological questions which were part of my questions and those of the other reviewers. Still, this is the authors' paper, not mine. The revised manuscript is very nice.

Response to Referees

We would like to thank all the referees for exemplary service. Their reports were extremely thorough and helpful, and have greatly improved the paper. As you can see below, Referees 1, 2 and 4 are satisfied with the current version of the MS.

Reviewer #1 (Remarks to the Author):

The authors addressed the comments to my satisfaction.

Reviewer #2 (Remarks to the Author):

The revised version is deal with my previous questions. I am happy to recommend it published as present version.

Reviewer #4 (Remarks to the Author):

The authors have responded to the comments well, and I believe that the manuscript can be accepted at this point. I still feel that it was overkill to have three different types of models, which the authors say is necessary to demonstrate universality to a biology-focused audience, but I think that this could have been done in a citation to the huge literature, leaving more space for the biological questions which were part of my questions and those of the other reviewers. Still, this is the authors' paper, not mine. The revised manuscript is very nice.

Referee 3, however, had three remaining questions.

Reviewer #3 (Remarks to the Author):

Question 1:

This new material on this theme is good. Entirely at their discretion, the authors might also consider citing Rendueles, Amherd & Velicer 2015, (Current Biology 25: 1673–1681) when they cite previous work that has proposed that microbial antagonisms might promote cooperation and when they discuss the potential for such antagonisms to generate positive frequency dependence (PFD) of fitness that in turn purges local within-patch diversity while generating inter-patch spatial structure that could promote cooperation. In that study, Rendueles et al. experimentally demonstrated the occurrence of PFD due to inter-strain antagonisms that the current authors propose hypothetically. Rendueles et al. also showed that PFD can simultaneously purge local within-patch diversity while promoting inter-patch diversity across a larger spatial scale and suggested that such spatial structure generated by PFD antagonisms (including any caused by Type VI secretion systems) could promote cooperation.

We now cite this paper when discussing frequency dependence- thank you for the excellent suggestion (and referral to an interesting paper!).

Question 2:

I find the manuscript greatly improved by the careful presentation of multiple possible explanations of the observed correlation. Regarding the fourth hypothesis, I would argue that it is more plausible than the authors suppose, because it would seem much easier to complexify a microbial cell's exobiochemistry without negative fitness effects than that of its inner workings. I would guess that a larger proportion of foreign genes taken in by transformation (or whatever means) that affect a cell's exome would be neutral or beneficial than of foreign genes encoding products that remain inside the cell. However, because the authors present the fourth hypothesis straightforwardly, I don't think any change to the text needs to be made on this point.

Sounds good!

Question 3:

In their rebuttal, the authors write "even antagonistic secretions can be considered cooperative from the producer genotype's perspective."

Yes, this is true. But the directionality of antagonism vs. cooperation is key here. The authors are proposing that a T6SS suite in species A could promote the stability of a distinct extracellular Secretion X by Species A because it prevents Species B from exploiting, i.e. benefiting from, Secretion X. But if Secretion X is antagonistic toward Species B rather than beneficial (like the T6SS of Species A), Species B is inherently unable to exploit Secretion X. In this scenario, the T6SS suite of Species A inherently does not protect producers of Secretion X from exploitation by Species B because the latter are harmed rather than helped by Secretion X. The only cells that could exploit antagonistic Secretion X are non-producers of X that remain resistant to X and that presumably share the same T6SS suite as producers, thus making T6SS identity irrelevant to the dynamics of cooperation and exploitation with respect to Secretion X.

To put this more succinctly, when a T6SS suite and a distinct secretion (say X) are both antagonistic toward the same target, neither protects the other from exploitation by that target. The point of my previous comment was that some secretions encoded in the same genome as a given T6SS suite might antagonize in the same direction at the T6SSs, thus eliminating them from the set of secretions that T6SSs might protect from exploitation. I think the authors should acknowledge this point. Whether this scenario applies to a substantial fraction of a given genome's exomic genes would need to be determined empirically.

This is a somewhat confusing question, but the argument is well-summarized here: "*when a T6SS suite and a distinct secretion (say X) are both antagonistic toward the same target, neither protects the other from exploitation by that target.*" We don't think this is an issue, however, for the following reasons: 1) If both the T6SS and distinct secretion produced by strain A kill the same target (strain B) then if anything we would expect a negative correlation between T6SS and other secretions as they would represent alternative killing strategies and this is not what we find empirically, and 2) Whenever a secretion kills a third competitor (strain C) it can clearly be seen as a cooperative trait benefitting strain A and strain B, and thus is captured in essence by our model in Figure 3 from which our predictions follow.

However, we do think that the antagonistic secretions are likely to play an important role structuring microbial communities. We haven't done the modeling or experiments yet (though we plan to), but we expect that diffusible antibiotics will also drive phase separation, acting synergistically with the T6SS to structure microbial communities. We have modified the discussion to bring this to the reader's attention:

While we restrict our analysis in this paper to the T6SS, the role of antagonistic interactions in structuring biological communities is probably far more general, applying to diffusible compounds that kill adjacent non-kin in both micro-organisms (*e.g.*, antibiotics) and macro-organisms (*e.g.*, allelopathy in plants⁵⁸ and animals⁵⁹).